# Antipsychotic-induced epigenomic reorganization in frontal cortex of individuals with schizophrenia

**Bohan Zhu[1†], Richard I Ainsworth[2†‡], Zengmiao Wang[2], Zhengzhi Liu[3], Salvador Sierra[4§], Chengyu Deng[1], Luis F Callado[5], J Javier Meana[5], Wei Wang[2,6]*, Chang Lu[1]*, Javier González-Maeso[4]***

[1]Department of Chemical Engineering, Virginia Tech, Blacksburg, United States; [2]Department of Chemistry and Biochemistry, University of California, San Diego, La Jolla, United States; [3]Department of Biomedical Engineering and Mechanics, Virginia Tech, Blacksburg, United States; [4]Department of Physiology and Biophysics, Virginia Commonwealth University School of Medicine, Richmond, United States; [5]Department of Pharmacology, University of the Basque Country UPV/EHU, CIBERSAM, Biocruces Health Research Institute, Bizkaia, Spain; [6]Department of Cellular and Molecular Medicine, University of California, San Diego, La Jolla, United States

***For correspondence:**
wei-wang@ucsd.edu (WW);
changlu@vt.edu (CL);
javier.maeso@vcuhealth.org (JG-M)

[†]These authors contributed equally to this work

**Present address:** [‡]Department of Medicine, Kao Autoimmunity Institute, Cedars-Sinai Medical Center, Los Angeles, United States; [§]Department of Neurology, University of Michigan, Ann Arbor, Ann Arbor, United States







**Abstract** Genome-wide association studies have revealed >270 loci associated with schizophrenia risk, yet these genetic factors do not seem to be sufficient to fully explain the molecular determinants behind this psychiatric condition. Epigenetic marks such as post-translational histone modifications remain largely plastic during development and adulthood, allowing a dynamic impact of environmental factors, including antipsychotic medications, on access to genes and regulatory elements. However, few studies so far have profiled cell-specific genome-wide histone modifications in postmortem brain samples from schizophrenia subjects, or the effect of antipsychotic treatment on such epigenetic marks. Here, we conducted ChIP-seq analyses focusing on histone marks indicative of active enhancers (H3K27ac) and active promoters (H3K4me3), alongside RNA-seq, using frontal cortex samples from antipsychotic-free (AF) and antipsychotic-treated (AT) individuals with schizophrenia, as well as individually matched controls (n=58). Schizophrenia subjects exhibited thousands of neuronal and non-neuronal epigenetic differences at regions that included several susceptibility genetic loci, such as *NRG1*, *DISC1,* and *DRD3*. By analyzing the AF and AT cohorts separately, we identified schizophrenia-associated alterations in specific transcription factors, their regulatees, and epigenomic and transcriptomic features that were reversed by antipsychotic treatment; as well as those that represented a consequence of antipsychotic medication rather than a hallmark of schizophrenia in postmortem human brain samples. Notably, we also found that the effect of age on epigenomic landscapes was more pronounced in frontal cortex of AT-schizophrenics, as compared to AF-schizophrenics and controls. Together, these data provide important evidence of epigenetic alterations in the frontal cortex of individuals with schizophrenia, and remark for the first time on the impact of age and antipsychotic treatment on chromatin organization.

## eLife assessment

The study by Zhu et al. provides **important** insights into cell-specific genome-wide histone modifications in the frontal cortex of individuals with schizophrenia, as well as shedding light on the role of

age and antipsychotic treatment in these associations. The evidence supporting the conclusions is **solid**.

## Introduction

Schizophrenia has traditionally been viewed as a genetic disorder, with heritability rates estimated at ~73% (***Sawa and Snyder, 2002***; ***Freedman, 2003***). However, previous genome-wide association studies (GWAS) clearly showed a relatively low number of genetic regions associated with schizophrenia risk – these include 108 loci in the first study (***Schizophrenia Working Group of the Psychiatric Genomics Consortium, 2014***) that have been expanded to over 270 regions (***Farrell et al., 2015***; ***Trubetskoy et al., 2022***). Most of these genetic variants were located in non-coding regions and hence, with a few exceptions there is little evidence supporting that coding variants contribute to schizophrenia risk, which also suggests that genetic factors do not seem to be sufficient to fully explain the molecular causes underlying this severe psychiatric condition.

Twin studies have provided evidence that environmental factors contribute to schizophrenia susceptibility. Thus, it had originally been suggested that monozygotic twins, whose DNA sequences are approximately 100% identical, have a concordance for schizophrenia of nearly 50% (***Cardno and Gottesman, 2000***; ***Hallmayer et al., 2011***), yet recent research has revealed a probandwise concordance rate of 33% in monozygotic twins and 7% in dizygotic twins (***Hilker et al., 2018***). While these findings underscore the substantial influence of genetic factors on the etiology of schizophrenia, they also advocate for a significant involvement of environmental events in the intricate development of this complex disorder (***Li et al., 2021***). This concept is further supported by epidemiological studies suggesting that prenatal environmental insults, such as maternal infection (***Brown et al., 2004***; ***Yudofsky, 2009***) and severe adverse life events (***Malaspina et al., 2008***), increase the risk of schizophrenia in the offspring.

Gene expression is regulated by the ability of the transcriptional machinery to access DNA, which is tightly packed into chromatin. The status of chromatin organization depends on epigenetic factors, such as DNA methylation and histone modifications that primarily occur on amino-terminal tails (***Gräff and Tsai, 2013***; ***Onuchic et al., 2018***; ***Bastle and Maze, 2019***). Hence, these epigenetic mechanisms lead to stable changes in gene expression that are mediated via altered chromatin structure without modification of DNA sequence, and remain largely plastic throughout all periods of brain development and aging. It is then tempting to speculate that epigenetic mechanisms mediate, at least in part, the effects of environmental factors on central nervous system (CNS) gene activity, and are, therefore, potentially involved in the pathophysiology of schizophrenia and other mental illnesses.

Supporting this concept, previous studies reported alterations in chromatin structure and accessibility in tissue samples from schizophrenic subjects and controls. Most of these previous reports, however, were focused on DNA methylation differences in peripheral blood (***Aberg et al., 2014***) and brain (***Jaffe et al., 2016***; ***Mendizabal et al., 2019***). Using the assay for transposase-accessible chromatin sequencing (ATAC-seq) in bulk tissue homogenates of postmortem frontal cortex samples, only a few differences in chromatin accessibility were observed between schizophrenia subjects and controls, in contrast to thousands of age-related differential accessible chromatin regions (***Bryois et al., 2018***). Histone modifications, including histone H3 acetylation of lysine 27 (H3K27ac) and histone H3 trimethylation of lysine 4 (H3K4me3) are critically involved in epigenomic regulations; H3K27ac marks active enhancers (***Creyghton et al., 2010***), whereas H3K4me3 marks active promoters (***Bernstein et al., 2005***). Enhancers are highly dynamic cis-regulatory elements with known involvement in neurodevelopmental processes (***Won et al., 2019***), and the dynamics of promoters are also significantly connected with the genetic risk of certain psychiatric conditions (***Dincer et al., 2015***). However, very few studies have been conducted about potential cell-type-specific genome-wide variations in covalent histone modifications in postmortem human brain samples of individuals with schizophrenia.

As an example, recent work combined fluorescence-activated cell sorting (FACS) of neuronal and non-neuronal cell nuclei with chromatin immunoprecipitation sequencing (ChIP-seq) assays in two brain regions (prefrontal cortex and anterior cingulate cortex) from postmortem brain samples of subjects without any known neurological or psychiatric disease (***Girdhar et al., 2018***). Besides the identification of cell and region-specific histone modification landscapes in this cohort of control subjects, these findings also compared their datasets with previous GWAS of individuals with psychiatric conditions,

reporting that strong specific enrichments occurred with schizophrenia and weaker associations with depression in both H3K27ac and H3K4me3 peaks. This correlation was almost exclusively observed in neuronal chromatin, but not in non-neuronal cell nuclei. More recent investigation conducting H3K27ac and H3K4me3 ChIP-seq assays in cortical neurons from schizophrenia subjects and controls has identified rare specific epigenetic variations for a set of non-coding RNA genes (***Gusev et al., 2019***) and chromatin domain alterations (***Girdhar et al., 2022***) that may contribute to the pathogenesis of schizophrenia. However, these previous epigenomic studies in postmortem human brain samples did not address the potential effect of previous exposure to antipsychotics on the regulation of chromatin state. This is particularly relevant considering that repeated administration of antipsychotic medications leads to epigenetic modifications at selected gene regions in mouse (***de la Fuente Revenga et al., 2018***; ***Ma et al., 2018***) and postmortem human brain (***Kurita et al., 2012***; ***Ibi et al., 2017***) samples. Similarly, whether such type of schizophrenia-associated epigenomic changes are observable in non-neuronal frontal cortex nuclei remains unexplored; even though there is evidence that alterations in the glia may contribute to major psychiatric disorders (***Liu et al., 2022***).

Combining MOWChIP-seq (***Cao et al., 2015a***; ***Zhu et al., 2019***) and Smart-seq2 (***Picelli et al., 2013***) for low-input profiling of H3K27ac and H3K4me3 histone modifications and transcriptomes, respectively, here we present the first dataset with cell type-specific epigenomic and transcriptomic landscapes in postmortem frontal cortex samples from two cohorts of schizophrenics either previously treated or not with antipsychotic medications and control subjects individually matched by sex and age. Importantly, our analyses allow the identification of transcription factors (TFs), their regulatees, and genes that may be involved in either the therapeutic effects of antipsychotics or the cause of undesired antipsychotic-induced epigenomic aberrations.

# Results

## Quality assessment for ChIP-seq and RNA-seq datasets in frontal cortex from postmortem human brain samples

Frontal cortex is a brain region involved in processes affected in schizophrenia patients, such as perception, cognition, and sensorimotor gating (***Andreasen et al., 1994***). We selected bilateral frontal cortex (Brodmann area 9) gray matter from 58 brain samples (29 schizophrenia and 29 controls). Control subjects were individually matched based on sex and age, and to a lower degree on postmortem delay (or PMD – time between death and tissue sample collection) (***Supplementary files 1 and 2***). Nuclei were FACS-sorted using an anti-NeuN antibody as a marker of neuronal nuclei, and NeuN-positive ($NeuN^+$) and NeuN-negative ($NeuN^-$) nuclei (approximately 60,000 nuclei per $NeuN^+$ or $NeuN^-$ sample) were collected for ChIP-seq (10,000 nuclei per library) and RNA-seq (6000 nuclei per library) (***Figure 1A***). After library preparation and sequencing, our MOWChIP-seq technology generated high-quality ChIP-seq data with average unique reads of ~11 million and ~14 million on histone modifications H3K27ac and H3K4me3, respectively (***Figure 1—figure supplement 1A***; and ***Supplementary file 3***). These yields were comparable to those in our previous studies using mouse frontal cortex and mammalian tissue culture samples (***Zhu et al., 2019***; ***de la Fuente Revenga et al., 2021***). We generated saturation curves to validate that our sequencing depth is sufficient, and that a further increase in the sequencing depth would not lead to significantly more called peaks (***Figure 1—figure supplement 2***). MOWChIP-seq datasets have very low background noise (***de la Fuente Revenga et al., 2021***) with the fraction of reads in called peaks (FRiP) average at 17.35% and 27.59% for our H3K27ac and H3K4me3 profiling, respectively (***Figure 1—figure supplement 1B***; and ***Supplementary file 3***). The PCR bottleneck coefficient (PBC) was calculated to measure library complexity (0.90 and 0.92 for H3K27ac and H3K4me3, respectively), which indicates that most of our ChIP-seq datasets have no or mild bottlenecking (***Figure 1—figure supplement 1C***; and ***Supplementary file 3***).

Using *Phantompeakqualtools* (***Marinov et al., 2014***), we calculated the normalized strand cross-correlation (NSC) and relative strand cross-correlation (RSC) to demonstrate the enrichment of sequencing reads around the histone modification sites (***Figure 1—figure supplement 1D and E***; and ***Supplementary file 3***). The average NSC was 1.15 and 1.23 for H3K27ac and H3K4me3, respectively; and the average RSC was 3.75 and 2.09 for H3K27ac and H3K4me3, respectively. These NSC and RSC values were higher than the recommended thresholds of 1.05 and 1.0, respectively. We also compared our GC metrics with those of previously published ChIP-seq data in postmortem human frontal cortex

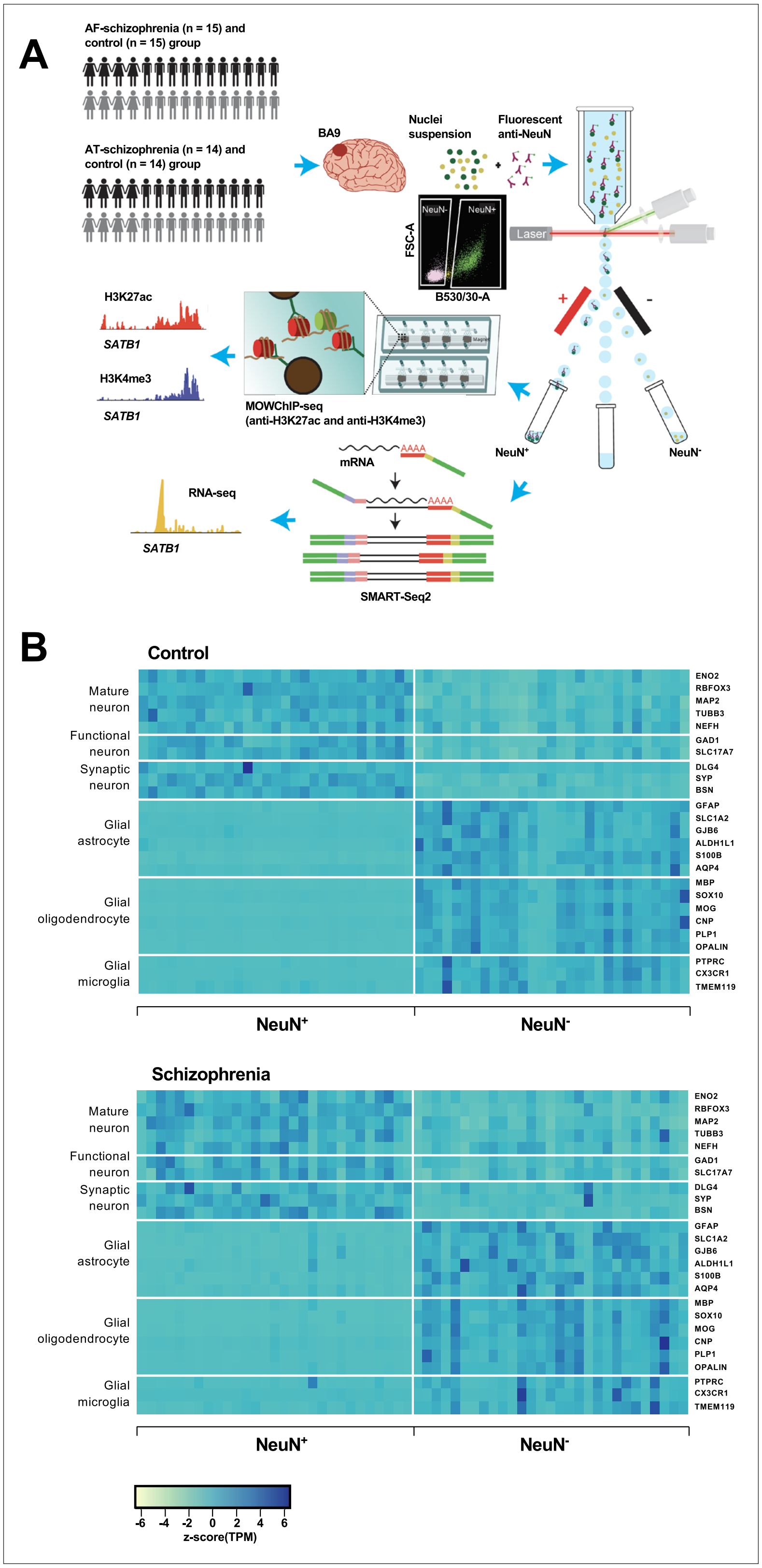


**Figure 1.** Overview of the multi-omics protocol for analyzing frontal cortex of schizophrenia subjects and controls. (**A**) Overview of the experimental design starting from postmortem human frontal cortex samples to generate cell type-specific H3K27ac, H3K4me3, and RNA profiles. (**B**) Heatmap of the expression of neuronal and glial cell markers across all NeuN$^{+}$ and NeuN$^{-}$ frontal cortex samples from 29 control subjects and 29 schizophrenia subjects.

*Figure 1 continued*

The online version of this article includes the following figure supplement(s) for figure 1:

**Figure supplement 1.** Distribution of six quality control metrics for ChIP-seq data on H3K4me3 and H3K27ac in NeuN-positive (NeuN$^+$) and NeuN-negative (NeuN$^-$) nuclei from schizophrenia and control groups, respectively.

**Figure supplement 2.** Saturation curves on the relationship between the number of sequencing reads and the number of identified peaks with our ChIP-seq data on H3K4me3 and H3K27ac.

**Figure supplement 3.** Distribution of four quality control metrics for RNA-seq data in NeuN-positive (NeuN$^+$) and NeuN-negative (NeuN$^-$) nuclei from schizophrenia and control groups.

**Figure supplement 4.** Representative results from fluorescence-activated cell sorting (FACS) sorting demonstrating the separation of neuronal (NeuN$^+$) and non-neuronal (NeuN$^-$) nuclei using fluorescence-labeled anti-NeuN antibody in postmortem human frontal cortex samples.

**Figure supplement 5.** Expression of neuronal (NeuN$^+$) and non-neuronal (NeuN$^-$) representative marker genes.

**Figure supplement 6.** Venn diagrams of the overlap between the identified peaks from our ChIP-seq study (green) and previous datasets (***Girdhar et al., 2022***) (cyan).

**Figure supplement 7.** Visualization of peak-wise or gene-wise means and variances of ChIP-seq and RNA-seq data, respectively, by voom plots.

samples (***Girdhar et al., 2018***). This previous study had an average NSC of 1.33 and 2.81, and RSC of 1.18 and 1.06 for H3K27ac and H3K4me3, respectively. The average Pearson's correlations between replicates were 0.923 and 0.971 for H3K27ac and H3K4me3, respectively, which compare well with those of ENCODE data (***ENCODE Project Consortium, 2012***). Our RNA-seq datasets had an average of ~6.8 million uniquely mapped reads, and the average mapping rate was 82.5% (***Figure 1—figure supplement 3***; and ***Supplementary file 4***), within the recommended range of 70–90% (***Conesa et al., 2016***). Our average GC content was 40.73% and exon percentage was 36.52%.

Using all frontal cortex samples from the 29 schizophrenia subjects and 29 controls, we analyzed the expression of selected neuronal and non-neuronal marker genes. Highly significant (median p-value = $6\times10^{-7}$) pair-wise differences in molecular marker expression were observed for all markers ranging from mature, functional, and synaptic neuron markers to astrocyte, oligodendrocyte, and microglial markers (***Figure 1B***; ***Figure 1—figure supplements 4 and 5***; ***Supplementary file 5***) – confirming neuronal and non-neuronal cell-type identities in the NeuN$^+$ and NeuN$^-$ nuclei samples, respectively.

Consequently, it can be concluded that MOWChIP-seq technology offers data quality comparable to that of state-of-the-art standard reference epigenomes while allowing histone modification profiling using small and highly purified populations of neuronal and non-neuronal nuclei from postmortem human frontal cortex samples.

## Epigenome and transcriptome profiling in neuronal and non-neuronal nuclei from frontal cortex of schizophrenia subjects and controls

To gain insight into the cell type-specific epigenomic changes associated with schizophrenia, we profiled histone marks H3K27ac and H3K4me3 and transcriptomes in NeuN$^+$ and NeuN$^-$ nuclei from the frontal cortex of schizophrenia subjects and controls. MOWChIP-seq data were mapped to the reference genome (Grch38), and significant peaks were called using MACS2 (***Figure 1—figure supplement 1F***). We then performed the overlap analysis to identify consensus peak sets for H3K27ac and H3K4me3 in NeuN$^+$ and NeuN$^-$ nuclei (see *Methods* section). We identified 107,938 consensus peaks covering ~135 Mb (4.22% of the genome) for H3K27ac in NeuN$^+$ nuclei, 71,490 consensus peaks covering ~101 Mb (3.18% of the genome) for H3K27ac in NeuN$^-$ nuclei, 13,3137 consensus peaks covering ~164 Mb (5.12% of the genome) for H3K4me3 in NeuN$^+$ nuclei, and 100,745 consensus peaks covering ~142 Mb (4.43% of the genome) for H3K4me3 in NeuN$^-$ nuclei. This peak distribution was consistent with previous H3K27ac and H3K4me3 NeuN$^+$ ChIP-seq studies in postmortem human frontal cortex tissue samples (***Girdhar et al., 2022***; ***Figure 1—figure supplement 6***). Voom plots using raw binding affinity matrix in ChIP-seq and RNA-seq datasets validate that low-enriched/expressed peaks/genes were filtered before downstream analysis (***Figure 1—figure supplement 7***). This is further corroborated with the smoothly decreasing curves fitted to the square root of residual standard deviation by average expression in all cases (***Figure 1—figure supplement 7***).

We then investigated the differences in histone modification profiles and gene activity between the schizophrenia and control cohorts. We regressed demographic (sex, age at death, PMD, antemortem diagnosis) and technical (align rate, unique rate, FRiP, NSC, RSC, the number of identified peaks, and PBC) covariates (***Sun et al., 2016***). We defined differential H3K27ac peaks that have no overlap regions with promoters as differential enhancers. Our analysis revealed 2,301 differential enhancers, 262 differential promoters, and 802 differentially expressed genes (DEGs) in $NeuN^+$ nuclei between schizophrenia subjects and controls, while 2,657 differential enhancers, 360 differential promoters, and 1,043 DEGs were discovered in $NeuN^-$ nuclei (***Figure 2A***; ***Supplementary file 6***). We next leveraged the previously identified promoter-anchored chromatin loops in $NeuN^+$ and $NeuN^-$ nuclei to identify how differential enhancers are linked with genes (***Hu et al., 2021***). We successfully associated 639 and 714 cell-type-specific enhancers to 328 and 395 genes via enhancer-promoter interactions in $NeuN^+$ and $NeuN^-$ nuclei, respectively. The rest of the enhancers were associated with their nearest genes. We discovered that the schizophrenia group had varied H3K27ac/H3K4me3 levels and RNA-seq reads compared to controls at various loci involved in processes related to synaptic plasticity and cognitive processes or previously associated with schizophrenia risk (***Farrell et al., 2015***) – these included enhancer region of *NRG1* (*Neuregulin 1*) in $NeuN^+$ nuclei, enhancer region of *GRM3* (*Metabotropic glutamate 3 receptor*) in $NeuN^-$ nuclei, promoter region of *DRD3 (Dopamine D3 receptor*) in $NeuN^+$ nuclei, promoter region of *DISC1* in $NeuN^-$ nuclei, *CDK5* (*Cyclin-dependent kinase 5*) mRNA in $NeuN^+$ nuclei, and *GRIN2A (Glutamate ionotropic receptor NMDA type subunit 2 A*) mRNA in $NeuN^-$ nuclei (***Figure 2B and C***; ***Supplementary file 6***). The accuracy of H3K27ac and H3K4me3 differential peak calling between schizophrenia subjects and controls identified by MOWChIP-seq was validated by independent ChIP-qPCR analysis of selected loci (***Figure 2—figure supplement 1***).

We also constructed QQ plots to examine the validity of our differential analysis (***Figure 2—figure supplement 2***). The lambda values of our H3K4me3 ChIP-seq datasets were lower than 1 (i.e. 0.43 and 0.36 for $NeuN^+$ and $NeuN^-$ nuclei, respectively), suggesting that there are fewer differential promoters than in the normal distribution. This matches previous findings on postmortem human brain samples (***Girdhar et al., 2018***).

To further explore the association between differential epigenetic modifications and genetic loci previously associated with schizophrenia risk (***Trubetskoy et al., 2022***), we examined the overlap of differential enhancer and promoter peaks with genetic variants using linkage disequilibrium (LD) score regression (***Finucane et al., 2015***). Various other brain and non-brain-related traits were also considered for comparison (***Finucane et al., 2018***). Schizophrenia was the most significantly enriched trait for all the differential enhancer and promoter regions in $NeuN^+$ and $NeuN^-$ fractions (***Supplementary file 7***). Additionally, the level of enrichment was higher in differential enhancers as compared to promoters, and the highest in differential enhancers of $NeuN^+$ nuclei (***Supplementary file 7***).

To assess agreement with the literature, we compared the DEGs identified in our study with a previous single-nucleus RNA sequencing (snRNA-seq) study in the postmortem prefrontal cortex of schizophrenics and controls (***Ruzicka et al., 2020***). Importantly, 236 out of our 802 DEGs (p-value = $1.96\times10^{-11}$) in $NeuN^+$ nuclei, and 63 out of our 1,043 DEGs (p-value = $4.18\times10^{-6}$) in $NeuN^-$ nuclei were also identified in this previous single-cell dissection work. In $NeuN^+$ nuclei, several genes encoding metabotropic glutamate receptors (*GRM3, GRM5*) that are directly associated with schizophrenia risk (***Maj et al., 2016***) were found differentially expressed in both studies (***Supplementary file 8***). We also identified some novel genes, including *LRRTM3*, which regulates excitatory synapse development (***Um et al., 2016***), and *POU3F2*, which is viewed as a key regulator of gene expression in a schizophrenia-associated gene co-expression module (***Chen et al., 2018***; ***Supplementary file 8***).

We also overlapped genes identified from differential enhancers/promoters with DEGs from RNA-seq (***Figure 2D***; ***Supplementary file 9***). We found that 66 (p-value = $9\times10^{-3}$) and 148 (p-value = $1.7\times10^{-2}$) genes were identified in both the DEGs from RNA-seq and differential enhancers/promoters associated genes in $NeuN^+$ and $NeuN^-$ nuclei, respectively. Among these, several schizophrenia-associated genes were also detected, including *NTNG2*, which is known to be involved in neurodevelopmental disorders (***Dias et al., 2019***) and *GRIN3A*, a gene that encodes NMDA receptor subunits in neuronal nuclei (***Yu et al., 2018***).

For integrative analysis of these diverse epigenomic and transcriptomic data, we next employed an unbiased method called *EpiSig* (***Ai et al., 2018***), which combines the epigenomic and transcriptomic dataset into a single analysis to cluster regions with similar epigenomic profiles across all the $NeuN^+$

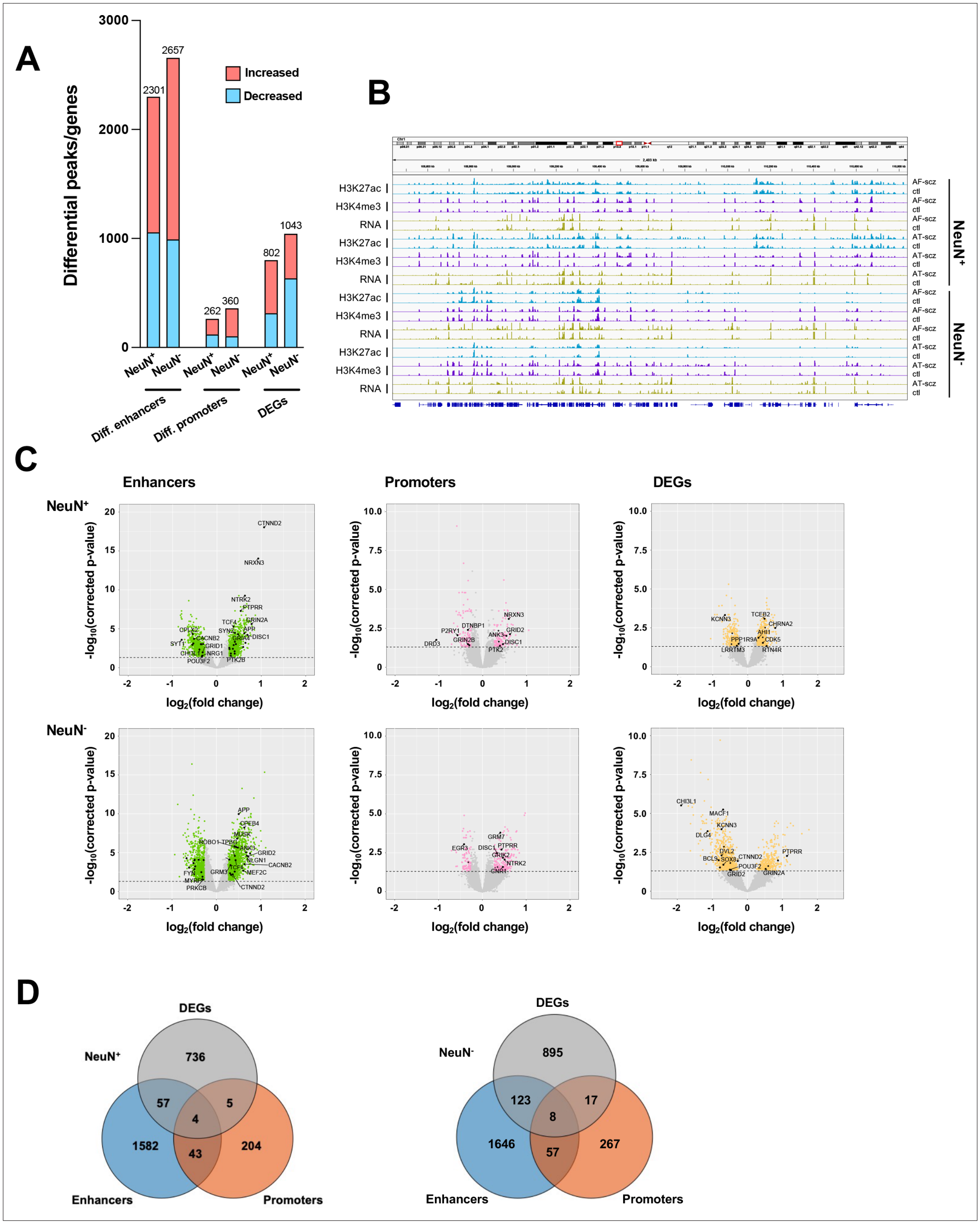


**Figure 2.** Comparison of epigenomic and transcriptomic landscapes in the frontal cortex of schizophrenia subjects and controls. (**A**) Differential enhancer/promoter peaks and differentially expressed genes (DEGs) obtained by comparing schizophrenia (n=29) and controls (n=29). The differential peaks or DEGs were identified using FDR <0.05. (**B**) Exemplar genomic track view of H3K27ac, H3K4me3, and RNA signals for matched AF-schizophrenia/control and AT-schizophrenia/control pairs in NeuN-positive (NeuN+) and NeuN-negative (NeuN-) cells. 50 Mb region displayed:

*Figure 2 continued*

chr1:68,000,000–118,000,000 (GRCh38). (**C**) Volcano plots showing genes associated with differential enhancer and promoter peaks and DEGs. Candidate genes for schizophrenia or genes involved in significant GO terms are labeled. The horizontal lines indicate FDR of 0.05. (**D**) Venn diagrams on the relationship among genes associated with differential enhancer or promoter peaks and DEGs.

The online version of this article includes the following figure supplement(s) for figure 2:

**Figure supplement 1.** qPCR validation of selected differential ChIP-seq peaks in $NeuN^+$ fraction for H3K27ac (**a-c**) and H3K4me3 (**d-f**).

**Figure supplement 2.** Q-Q plots of the corrected p-value from differential peaks at enhancer/promoter regions and differentially expressed genes.

and $NeuN^-$ nuclei samples. 85,462 signal-enriched regions were grouped into 814 epigenomic clusters covering 14.53% of the genome. These clusters were further combined into 6 groups (sections) using the K-means method (***Figure 3A and B***; ***Supplementary files 10 and 11***). Section I had high coverage in the gene annotations for intron (35%) and intergenic regions (29%) indicating inactive regions. It was also enriched in chromosome X compared to other sections. Section II was annotated as enhancers that are active in $NeuN^+$ nuclei but suppressed in $NeuN^-$ nuclei. A hypergeometric test identified clusters that were significantly enriched in schizophrenia vs control differential histone marks and differentially expressed genes (***Figure 3A and B***; ***Supplementary files 10 and 11***). The top five Section II clusters had schizophrenia-positive association (i.e. activity and expression schizophrenia >controls) in genes enriched in GO terms '*Trans-synaptic signaling*' (FDR $4.66\times10^{-2}$). Section III was highly enriched in enhancers (average of 21% of all regions in each cluster), and had low coverage in intergenic regions (12%), which is likely associated with active enhancers for both $NeuN^+$ and $NeuN^-$ given the high signal strength for H3K27ac. Top differentially enriched cluster genes, enriched in the GO term '*Amyloid fibril formation*' (FDR $5.99\times10^{-2}$), were found to be negatively associated with schizophrenia in $NeuN^+$ nuclei, whereas genes enriched in the GO term '*Neuron projection development*' (FDR $3.19\times10^{-2}$) were positively associated with schizophrenia in $NeuN^-$ nuclei (***Figure 3A and B***; ***Supplementary files 10 and 11***). Section IV also had high coverage of enhancers (***Figure 3A and B***; ***Supplementary files 10 and 11***). However, it had the highest average promoter content with 16% of all gene annotations being promoter regions, further supported by CpG islands showing the highest proportion (20%) in this section; indicating active promoters for both $NeuN^+$ and $NeuN^-$ nuclei. Both cell types showed enrichment in respiratory electron transport genes that were negatively associated with schizophrenia (***Figure 3A and B***; ***Supplementary files 10 and 11***). As for the average differential signals across sections, a great variance was observed. For example, the signal of H3K4me3 was higher in schizophrenia subjects compared to controls for $NeuN^+$ nuclei in section V, while it was lower in schizophrenia subjects for $NeuN^-$ samples (***Figure 3A and B***; ***Supplementary files 10 and 11***). Finally, Section VI was annotated as enhancers that were active in $NeuN^-$ nuclei but repressed in $NeuN^+$ nuclei (***Figure 3A and B***; ***Supplementary files 10 and 11***).

Transcriptional regulatory processes proceed as a hierarchy of orchestrated events that ultimately modulate the expression of downstream target genes. Using the recently developed *Taiji* algorithm (***Zhang et al., 2019***), which allows access to information pertaining to transcriptional cascades deriving from upstream drivers through specific pathway mechanisms to downstream effects, we integrated epigenomic and transcriptomic data to construct 116 individual transcriptional networks in neuronal and glial nuclei from schizophrenia subjects and controls. We identified active promoters and enhancers using H3K27ac and then predicted TF binding sites by scanning 1,165 TF motifs linking putative TF binding sites to their targets using *EpiTensor* (***Zhu et al., 2016***), an unsupervised method to predict enhancer-promoter associations. TFs were subsequently ranked according to regulatory importance using the Personalized PageRank (PPR) algorithm for each unique network topology (***Yu et al., 2017***). Using the differentially expressed TFs (schizophrenia *vs* controls FDR <0.05), TFs were ranked by absolute change in schizophrenia *vs* control PPR score (***Figure 4A*** – top 10 TFs for each cell type; ***Supplementary file 12***). Of the top 10 TFs of $NeuN^+$ nuclei, all were found to be cell-type specific. Using the top four TFs, we identified 207 regulatees the were regulated by three or more TFs and found they were involved in processes such as '*Neurexins and neuroligins*' (FDR $2.18\times10^{-7}$) and '*Protein-protein interactions at synapses*' (FDR $1.22\times10^{-6}$) (***Figure 4B***; ***Supplementary file 12***). Furthermore, all top 10 TFs of $NeuN^-$ nuclei were cell-type specific TFs and the regulatees of the top four TFs were enriched in signaling pathways including '*RAF/MAP kinase cascade*' (FDR $2.80\times10^{-2}$) and '*RHO GTPase cycle*' (FDR $3.07\times10^{-2}$) (***Figure 4C***; ***Supplementary file 12***).

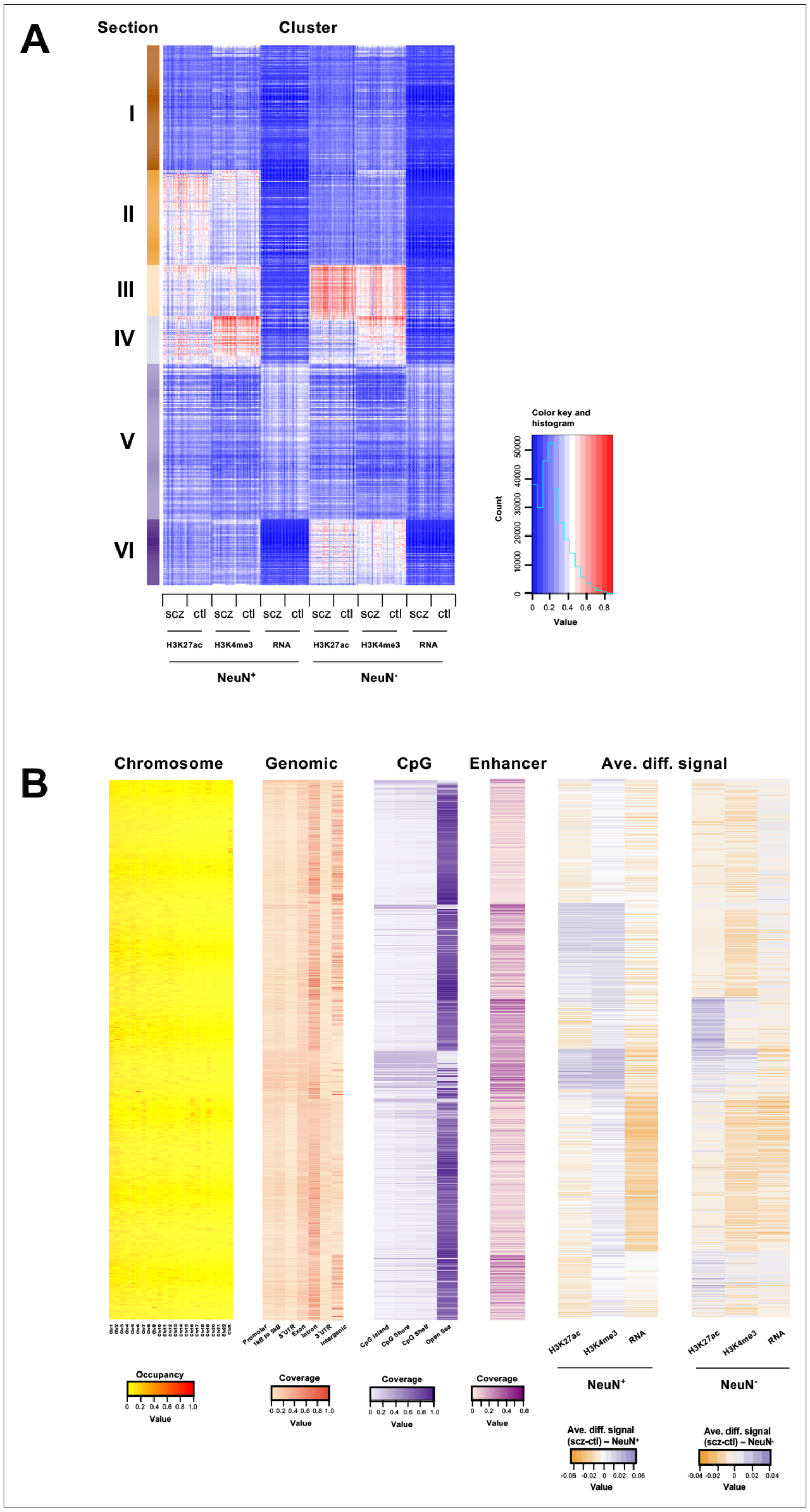


**Figure 3.** Genome-wide multidimensional clusters in the frontal cortex of schizophrenia subjects and controls. (**A**) Integrative analysis using *EpiSig*. 814 *EpiSig* clusters across 348 genome-wide sequencing datasets were grouped into 6 sections. The heatmap shows the signal in each *EpiSig* cluster (row: *EpiSig* cluster; column: marker). (**B**) For each *EpiSig* cluster, from left to right, the heatmaps are: the region percentage in each chromosome; the genomic annotation; the CpG annotation; the percentage of enhancer; the difference signal between schizophrenia and controls in NeuN-positive (NeuN$^+$) and NeuN-negative (NeuN$^-$) nuclei.

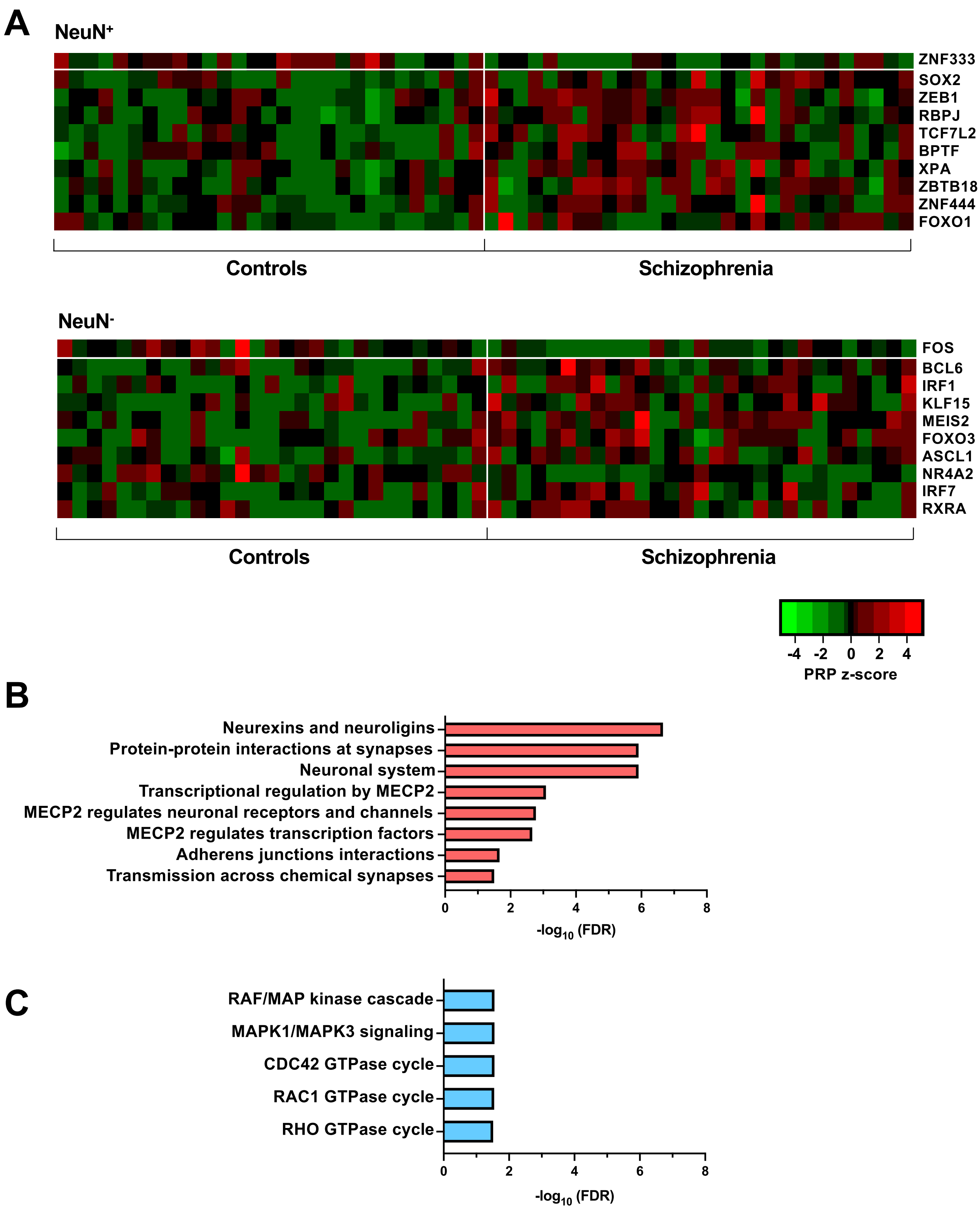


**Figure 4.** Transcriptional regulatory processes in the frontal cortex of schizophrenia subjects and controls. (**A**) Heatmap of z-score Personalized PageRank (PPR) for top 10 significantly differentially expressed transcription factors (TFs) (FDR <0.05) ranked by absolute change in PPR for NeuN-positive (NeuN$^+$) (upper panel) and NeuN-negative (NeuN$^-$) (lower panel) schizophrenia *vs* control nuclei samples. (**B**) Overrepresented pathway analysis (FDR <0.05) for 203 downstream regulatees common to the top four schizophrenia *vs* control NeuN$^+$-specific transcription factors (TFs) (*ZNF333*, *SOX2*,

*Figure 4 continued*

*ZEB1*, and *RBPJ*). (**C**) Overrepresented pathway analysis (FDR <0.05) for 225 downstream regulatees common to the top four schizophrenia *vs* control NeuN$^-$-specific TFs (*FOS*, *BCL6*, *IRF1*, and *KLF15*).

## Alterations in antipsychotic-free but not in antipsychotic-treated schizophrenics

Using preclinical models, it has been suggested that chronic antipsychotic drug administration leads to long-lasting changes in frontal gene expression and chromatin organization (**Kurita et al., 2012**; **de la Fuente Revenga et al., 2018**), but the epigenomic consequences of antipsychotic treatment in postmortem human brain samples remain largely unexplored. To validate the separation between the antipsychotic-free (AF) schizophrenia and antipsychotic-treated (AT) schizophrenia groups, we first utilized a dimension reduction algorithm – uniform manifold approximation and projection (UMAP) – to visualize the clustering of each sample with TMM normalized binding affinity matrices or gene expression files from MOWChIP-seq and RNA-seq, respectively (**Figure 5—figure supplement 1**). The separation between AF-schizophrenia and AT-schizophrenia groups is clearly visible at enhancer and promoter regions for both NeuN$^+$ and NeuN$^-$ nuclei.

To further determine the functional relevance of antipsychotic treatment, we aimed to identify the biological pathways, TFs, or gene expressions dysregulated in the AF-schizophrenia group that were also reversed to control levels in the AT-schizophrenia group as compared to individually matched controls. Hence, these represent schizophrenia-associated molecular alterations that are reversed upon antipsychotic treatment. We calculated the average pairwise difference in PageRank in NeuN$^+$ nuclei from the AF-schizophrenia/control pair cohort. In conjunction with this, we also calculated the average pairwise difference in PageRank in NeuN$^+$ nuclei from the AT-schizophrenia/control pair cohort. We then identified those TFs with a difference in these two values greater than 0.5 (**Figure 5A**; **Supplementary file 13**). However, when these TFs were further filtered based on a significance cut-off of FDR <0.05, no significant TFs were identified to be simultaneously changed in the AF-schizophrenia/control group and not changed in the AT-schizophrenia/control group (**Figure 5C**; **Supplementary file 13**). In parallel, AF-schizophrenia/control DEGs were integrated with genes from the AT-schizophrenia/control cohort with no case/control differences in expression resulting in a list of 116 cohort-specific DEGs (**Figure 5C**; **Supplementary file 13**). Functional enrichment analysis of these genes resulted in pathways involved in glutamatergic neurotransmission including '*Activation of AMPK downstream of NMDARs*' (FDR $3.66\times10^{-3}$) (**Figure 5G**; **Supplementary file 13**). Structural and functional modifications of dendritic spines are central to brain development and plasticity (**Spruston, 2008**). Studies from postmortem brains of subjects with neurodevelopmental disorders including schizophrenia demonstrate altered density and morphology of dendritic spines, particularly in the frontal cortex (**Glantz and Lewis, 2000**; **Black et al., 2004**). IQGAP scaffold proteins facilitate the formation of complexes that regulate cytoskeletal dynamics including microtubules (**Cao et al., 2015b**). Interestingly, another significant pathway restored in the AT-schizophrenia group was '*Rho GTPases activate IQGAPs*' (FDR $3.66\times10^{-3}$) (**Figure 5G**; **Supplementary file 13**). The importance of this pathway was validated by the analysis of the clusters from the *EpiSig* pipeline. Thus, taking the top three clusters ranked for enrichment in H3K27ac and mapping their differential peaks to genes resulted in 166 genes enriched in pathways including '*Adherens junctions interactions*' (p-value $1.22\times10^{-4}$) (**Figure 5—figure supplement 2**; **Supplementary file 14**).

Autophagy has been suggested to play an important role in the pathophysiology of schizophrenia and antipsychotics are known to modulate the process (**Merenlender-Wagner et al., 2015**). Notably, the pathways '*Aggrephagy*' (FDR $3.66\times10^{-3}$) and '*Macroautophagy*' (FDR $7.71\times10^{-3}$) were significantly enriched (**Figure 5G**; **Supplementary file 13**). Expression of the '*Macroautophagy*' genes *AMBRA1*, *PRKAB1*, *TUBA1A*, *TUBB2A*, and *TUBA4A* was restored in the AT-schizophrenia group (**Figure 5I**; **Supplementary file 13**). Ubiquitin B (UBB) expression has been previously identified as a strong correlate of schizophrenia symptoms (**Rubio et al., 2013**). We show a 1.96-fold decrease specific to AF-schizophrenics compared to controls (p-value $=4.0\times10^{-2}$) (**Supplementary file 13**).

Glial cells modulate and act as effectors in neurodevelopment through a wide range of neuronal-glial cell interactions. Using the same process as above, we identified two driver TFs with a significant change in PPR between case and control for the AF-schizophrenia cohort and no significant

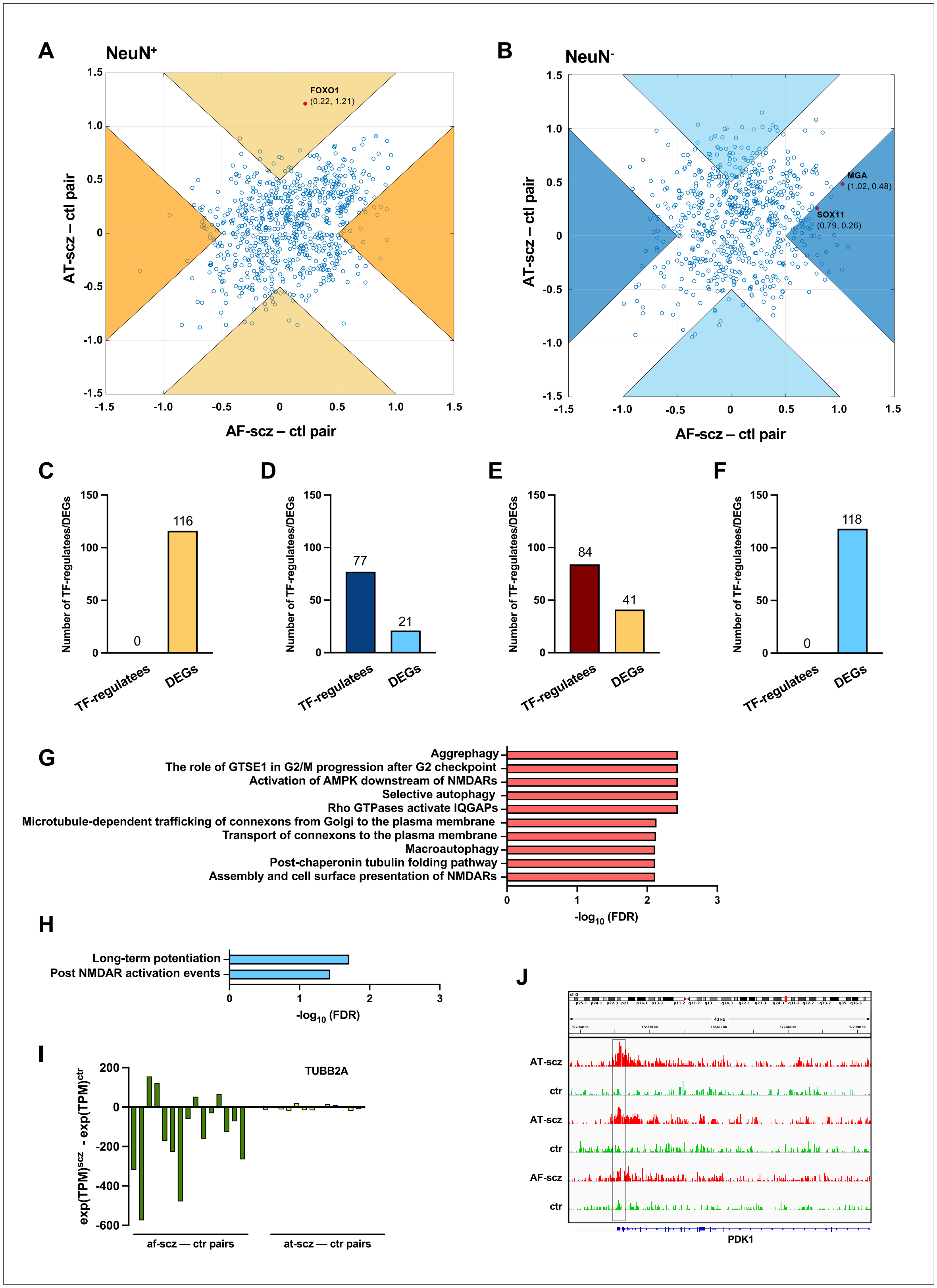


**Figure 5.** Epigenomic alterations affected by antipsychotic treatment. (**A**) Scatter plot of average pairwise change in Personalized PageRank (PPR) ($PPR_{schizophrenia}$ – $PPR_{control}$) for antipsychotic-free (AF) vs antipsychotic-treated (AT) NeuN-positive (NeuN+) cohorts. Orange regions show cohort (AF – AT)<0.5 (i.e. alterations recovered by antipsychotic treatment), whereas beige regions show cohort (AT – AF)>0.5 (i.e. alterations consequence of antipsychotic treatment). Transcription factors (TFs) FDR <0.05 highlighted in red. (**B**) Scatter plot of average pairwise change in PPR ($PPR_{schizophrenia}$ –

*Figure 5 continued*

$PPR_{control}$) for AF vs AT NeuN⁻ cohorts. Dark blue regions show cohort (AF – AT)<0.5 (i.e. alterations recovered by antipsychotic treatment), whereas cyan regions show cohort (AT – AF)>0.5 (i.e. alterations consequence of antipsychotic treatment). TFs FDR <0.05 highlighted in red. (**C**) Number of differentially expressed gene (DEG) regulatees by TFs, and number of DEGs in $NeuN^+$ nuclei from AF-schizophrenia/control pairs. (**D**) Number of DEG regulatees by TFs, and number of DEGs in $NeuN^-$ nuclei from AF-schizophrenia/control pairs. (**E**) Number of DEG regulatees by TFs, and number of DEGs in $NeuN^+$ nuclei from AT-schizophrenia/control pairs. (**F**) Number of DEG regulatees by TFs, and number of DEGs in $NeuN^-$ nuclei from AT-schizophrenia/control pairs. (**G**) Functional enrichment analysis of union of genes from AF-schizophrenia/control pairs in $NeuN^+$ nuclei. (**H**) Functional enrichment analysis of union of genes from AF-schizophrenia/control pairs in $NeuN^-$ nuclei. (**I**) Pairwise expression difference (schizophrenia – control) of an exemplar AF-schizophrenia/control cohort DEG (TUBB2A) across all 29 schizophrenia-control pairs in $NeuN^+$ nuclei. (**J**) H3K27ac tracks for PDK1 (member of the 84 gene set in E) in $NeuN^+$ nuclei. Box highlighting the FOXO1 DNA-binding motif in promoter at position chr2: 172,555,706–172,555,718 (GRCh38). Two exemplar AT-schizophrenia/control cohort pairs showing differential H3K27ac peak intensity around motif locus and an example AF-schizophrenia/control cohort patient pair with no difference.

The online version of this article includes the following figure supplement(s) for figure 5:

**Figure supplement 1.** Uniform manifold approximation and projection (UMAP) visualization of the feature matrix of enhancers, promoters, and RNA among antipsychotic-free (AF)-schizophrenia, antipsychotic-treated (AT)-schizophrenia, and control subjects.

**Figure supplement 2.** Differential peak enrichment analysis for antipsychotic-free NeuN-positive ($NeuN^+$) nuclei in *EpiSig* clusters.

**Figure supplement 3.** Differential peak enrichment analysis for antipsychotic-treated NeuN-positive ($NeuN^+$) nuclei in *EpiSig* clusters.

**Figure supplement 4.** Differential peak enrichment analysis for antipsychotic-treated NeuN-negative ($NeuN^-$) nuclei in *EpiSig* clusters.

difference between case and control in the AT-schizophrenia group, *SOX11* (FDR $1.73\times10^{-2}$) and *MGA* (FDR $9.83\times10^{-3}$) (***Figure 5B***; ***Supplementary file 13***). 77 downstream DEG regulatees of these two TFs were identified in the AF cohort showing significant regulatory case/control change (***Figure 5D***; ***Supplementary file 13***). In parallel, 21 cohort-specific DEGs were identified as AF-schizophrenia/control DEG and having no significant difference in expression in the AT-schizophrenia/control cohort (***Figure 5D***; ***Supplementary file 13***). Functional enrichment analysis of the union of 153 genes included '*Post NMDA receptor activation events*' (FDR $3.65\times10^{-2}$), and '*Long-term potentiation*' (FDR $1.95\times10^{-2}$) (***Figure 5H***; ***Supplementary file 13***). *EpiSig*'s analysis did not show $NeuN^-$ alterations in the AF-schizophrenia cohort.

We also performed the differential analysis with demographic and technical covariates regressed out on AF-schizophrenia/control and AT-schizophrenia/control cohorts. In $NeuN^+$ nuclei, the results revealed 2,069 and 574 differential enhancers and promoters, respectively, and 166 DEGs between AF-schizophrenia and their controls (***Figure 6A***; ***Supplementary file 15***), while 3,658, 36, and 1,273 differential enhancers, promoters, and DEGs were discovered between AT-schizophrenia and controls (***Figure 6B***; ***Supplementary file 15***). In $NeuN^-$ nuclei, we identified 891, 19, and 128 differential peaks/genes between AF-schizophrenia and controls (***Figure 6A***; ***Supplementary file 15***); 2,651, 775, 776 differential peaks/genes between AT-schizophrenia and controls, in enhancers, promoters, and DEGs, respectively (***Figure 6B***; ***Supplementary file 15***). More differential enhances/promoters and genes were detected between AT-schizophrenia and their matched controls than those between AF-schizophrenia and their controls with the exception in neuronal promoters (***Figure 6A and B***; ***Supplementary file 15***).

Similar to our TF analyses (***Figure 5***), we also identified the genes altered in the AF-schizophrenia/control group but not in the AT-schizophrenia/control group using differential analyses of enhancers, promoters, or gene expression. It should be noted that in the differential analyses here, the schizophrenia subjects (whether AF or AT) and their controls were compared at the cohort level, while matched schizophrenia/control pairs were examined individually in the TF-based analysis. At the epigenomic level, in $NeuN^+$ nuclei, we identified 687 and 549 genes changed in the AF- but not in AT-schizophrenics by examining differential enhancers and promoters, respectively (***Supplementary file 16***). These genes were linked to epigenomic features restored to their basal level after treatment. In $NeuN^-$ nuclei, there were 270 and 17 recovered genes linked with differential enhancers and promoters, respectively. At the transcriptomic level, 145 DEGs in $NeuN^+$ nuclei and 109 in $NeuN^-$ nuclei were discovered in AF-schizophrenia/control comparison but not in the AT-schizophrenia/control differential analysis.

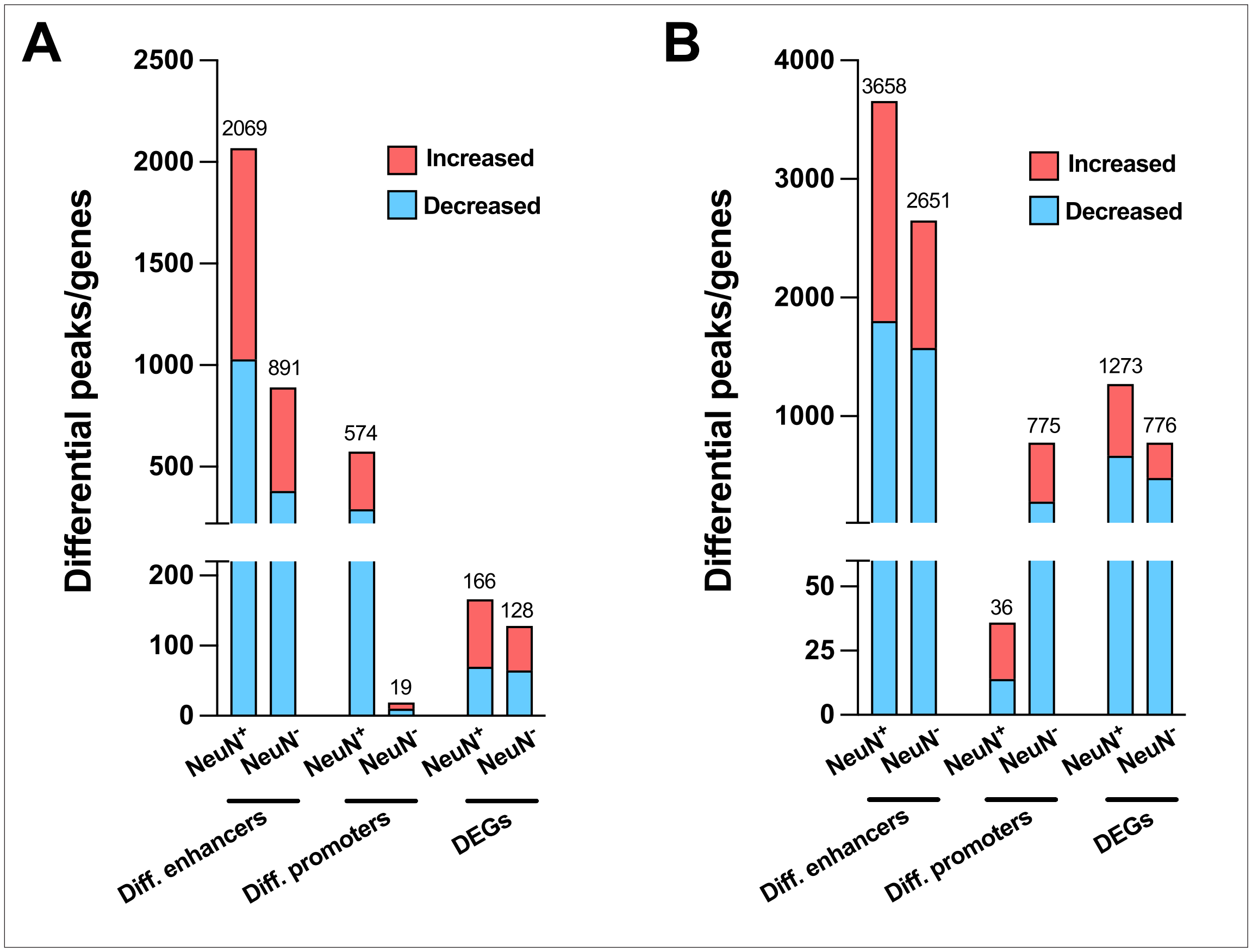


**Figure 6.** Effect of antipsychotic treatment on differential enhancers/promoters and differentially expressed genes (DEGs) in NeuN-positive (NeuN+) and NeuN-negative (NeuN-) nuclei from the frontal cortex of schizophrenia subjects and controls. (**A**) Differential enhancer/promoter peaks and DEGs were obtained by comparing antipsychotic-free (AF)-schizophrenics and individually matched controls. (**B**) Differential enhancer/promoter peaks and DEGs obtained by comparing antipsychotic-treated (AT)-schizophrenics and individually matched controls.

## Alterations in antipsychotic-treated but not in antipsychotic-free schizophrenics

We next sorted to characterize those TFs that exhibit regulatory alterations in the AT-schizophrenia/control cohort but not in the AF-schizophrenia/control cohort. Our goal was to identify modifications in pathways that represent a consequence of antipsychotic medication rather than an epigenetic mark of schizophrenia in the postmortem human brain (in other words, unwanted side effects caused by antipsychotic treatment). In the same way as above, we identified those TFs with a change in the AT-schizophrenia/control group but not in the AF-schizophrenia/control group for the NeuN+ nuclei (***Figure 5A***; ***Supplementary file 13***). Further filtering of these TFs based on a significance cut-off of FDR <0.05 leads to the identification of FOXO1 (FDR $4.89\times10^{-2}$).

We identified dysregulated AT-schizophrenia/control DEG regulatees of these TFs in NeuN+ nuclei via analysis of differential edge weights thus obtaining 84 genes (***Figure 5E***; ***Supplementary file 13***). AT-treated/control DEGs were intersected with genes from the AF-schizophrenia/control cohort with no case/control differences in expression resulting in a list of 41 cohort-specific DEGs (***Figure 5E***; ***Supplementary file 13***). Pathway analysis on the union of genes yields the pathway '*Regulation of p53 activity through phosphorylation*' (FDR $1.13\times10^{-2}$) (***Supplementary file 13***), including the FOXO1 AT-schizophrenia/control cohort DEG regulatees *CCNA1*, *BLM*, *TP53RK*, and *RBBP8*, and the AT-schizophrenia/control cohort-specific DEGs *PRKAA1* and *TAF15* (***Supplementary file 13***). The p53 regulatory gene PDK1 was also identified as a FOXO1 AT-schizophrenia/control DEG regulatee (***Figure 5J***; ***Supplementary file 13***). p53 is one of the most critical pro-apoptotic genes, and anti-psychotics are known to produce complex effects including the activation of both proapoptotic and antiapoptotic signaling pathways (***Aylon and Oren, 2007***). Our data showed that all genes involved

in the regulation of p53 presented a significantly lower expression in AT-schizophrenics compared to controls, suggestive of a repressive role for FOXO1 for its five regulatees, as FOXO1 had higher PPR in treated schizophrenics (**Supplementary file 13**) and was also 2.38-fold more highly expressed in schizophrenics for the treated-cohort (p-value = $1.07\times10^{-2}$) (**Supplementary file 13**). Analysis of the H3K4me3 enriched clusters from the *EpiSig* pipeline for the AT-schizophrenia/control cohort corroborated alterations in pathways related to p53 (**Endo et al., 2008**; **Figure 5—figure supplement 3**; **Supplementary file 17**).

In NeuN$^-$ nuclei, no TFs showed a significant difference in the AT-schizophrenia/control cohort but not in the AF-schizophrenia/control cohort (**Figure 5F**; **Supplementary file 13**). Furthermore, the 118 genes cohort-specific DEGs identified as AT-schizophrenia/control DEGs and having no significant difference in expression in the AF-schizophrenia/control cohort were not significantly enriched in any signaling pathway. Analysis of the clusters from the *EpiSig* pipeline remarked the importance of the RHO GTPase pathway on the regulatory alterations observed in AT-schizophrenia subjects (**Figure 5—figure supplement 4**; **Supplementary file 18**).

We also used differential analyses of enhancers, promoters, and expression to discover the genes altered in the AT-schizophrenia/control group but not in the AF-schizophrenia/control group. In NeuN$^+$ nuclei, we found 1,591 and 28 treatment-altered genes linked with differential enhancers and promoters, respectively (**Supplementary file 16**). In NeuN$^-$ nuclei, we identified 1,351 and 718 altered genes linked with differential enhancers and promoters, respectively (**Supplementary file 16**). At the transcriptomic level, 1,252 DEGs in NeuN$^+$ nuclei and 757 in NeuN$^-$ nuclei were discovered in AT-schizophrenia/control comparison but not in the AF-schizophrenia/control differential analysis.

## Age differentially affects antipsychotic-treated schizophrenia subjects

In order to further assess the effect of age on gene expression, we first compared transcriptomes of subjects with schizophrenia and the controls to evaluate how these changes correlated with age. Within NeuN$^+$ nuclei in the control group, we identified 742 genes that were significantly correlated with age – with most of them (573, or 77.2%) showing decreased expression in older control subjects (**Figure 7A**; **Supplementary file 19**). These included *APOL2*, which has been involved in epigenetic aging (**Luo et al., 2020**). The opposite, however, was observed in NeuN$^+$ nuclei from schizophrenia subjects with 18 out of 622 (2.8%) in AF-schizophrenia presenting a negative correlation with age, an effect that was partly reversed in the AT-schizophrenia cohort (85 out of 242 or 35.1%) (**Figure 7A**; **Supplementary file 19**).

Our data also demonstrate that within the NeuN$^-$ nuclei genes correlated with age (1,031, 389, and 351 in controls, AF-schizophrenia and AT-schizophrenia, respectively), approximately half (491 or 47.6%) were positively correlated with age in the control group whereas a much higher fraction of genes showed increased expression with age in the schizophrenia group, particularly in the AF-schizophrenia cohort (382 or 98.2% in AF-schizophrenia, and 276 or 78.6% in AT-schizophrenia) (**Figure 7B**; **Supplementary file 19**). These results suggest that age differentially affects gene expression in the frontal cortex of AF-schizophrenia vs AT-schizophrenia subjects as compared to age-matched controls. Importantly, this was further confirmed by the functional integration of epigenomic and transcriptomic data and the evaluation of how these alterations correlated with age.

Thus, we evaluated pairwise changes in expression between schizophrenia subjects and their age-matched controls, and identified 206 and 310 genes with an absolute Pearson's correlation of ≥0.50 in NeuN$^+$ nuclei from the AF-schizophrenia/control and AT-schizophrenia/control cohorts, respectively (**Figure 7C and D**; **Supplementary file 19**). We also found enriched biological processes associated with age, including '*Regulation of protein kinase activity*' (p-value $6.69\times10^{-7}$) in the AT-schizophrenia/control group (**Figure 7E**; **Supplementary file 19**). Within this gene set, the difference between AT-schizophrenia subjects and control pairs correlated either positively (*WNK1*) or negatively (*SFRP2*) with age (**Figure 7F and G**; **Supplementary file 19**). Evaluation of pairwise changes in PPR identified 48 TFs with high correlations to age in NeuN$^+$ nuclei from AT-schizophrenia/control cohorts (**Figure 7D**; **Supplementary file 19**), whereas this alteration was not observed in the AF-schizophrenia/control group (**Figure 7C**). Pathway analysis of the NeuN$^+$ TFs affected by age in the AT-schizophrenia/control cohort led to the top pathway '*NGF-simulated transcription*' (p-value $8.04\times10^{-8}$), including the TFs *EGR2* and *ATF2* (**Supplementary file 19**). Hallucinations and delusions typically attenuate with aging (**Davidson et al., 1995**), which is consistent with the lower PPR difference for *EGR2* – a preclinical

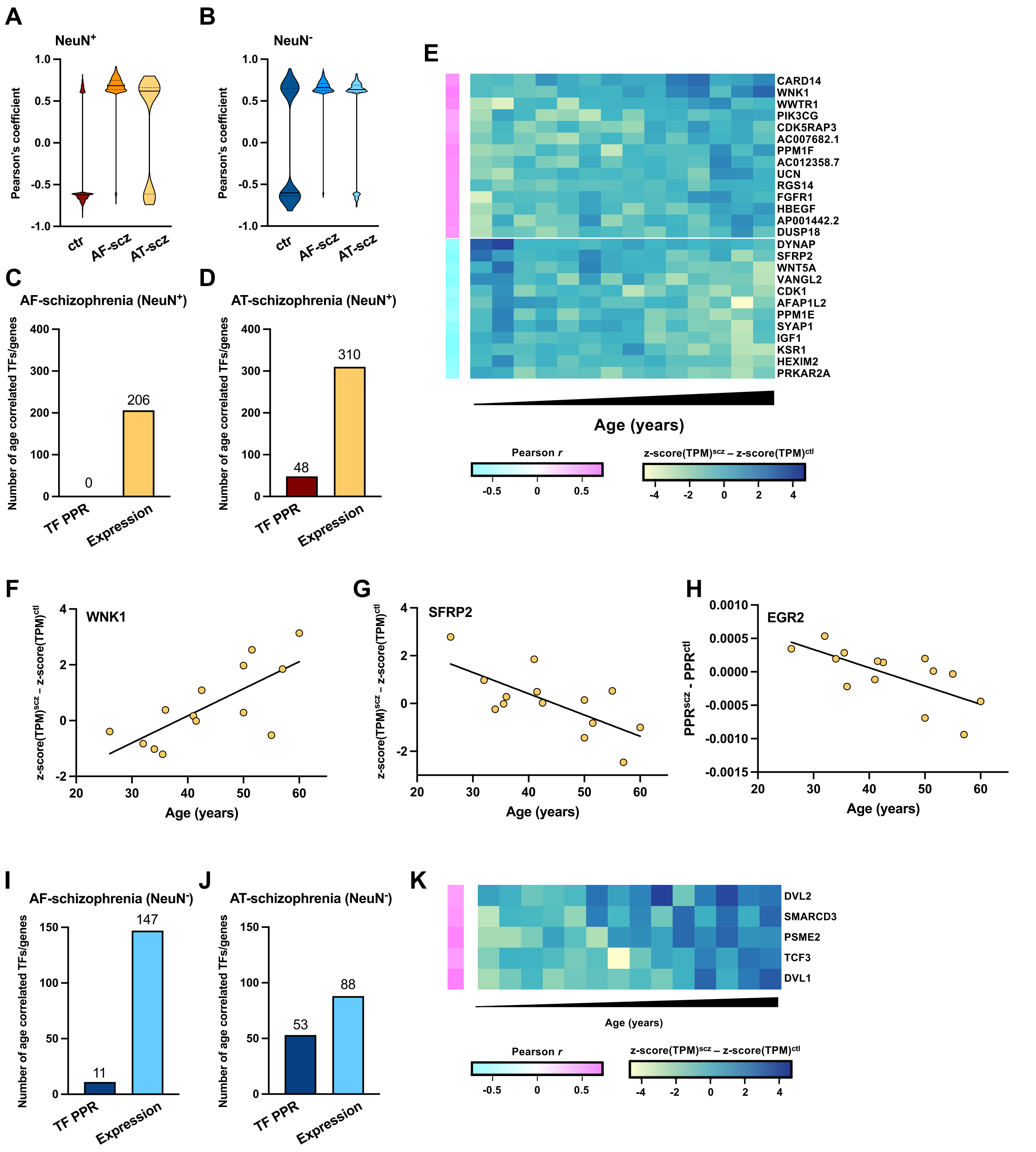


**Figure 7.** Epigenomic effect of age on treated schizophrenia subjects. (**A**) Violin plots for Pearson's R correlation coefficients of age *vs* expression f 742, 622, and 242 genes from control, antipsychotic-free (AF)-schizophrenia, and antipsychotic-treated (AT)-schizophrenia NeuN-positive (NeuN+) nuclei, respectively. (**B**) Violin plots for Pearson's R correlation coefficients of age *vs* expression for 1031, 389, and 351 genes from control, AF-schizophrenia, and AT-schizophrenia NeuN-negative (NeuN-) nuclei, respectively. (**C**) Number of pairwise transcription factor (TF) Personalized PageRank (PPR) and pairwise

*Figure 7 continued*

gene expression differences correlated with age in NeuN+ nuclei from AF-schizophrenia/control pairs. (**D**) Number of pairwise TF PPR and pairwise gene expression differences correlated with age in NeuN+ nuclei from AT-schizophrenia/control pairs. (**E**) Heatmap for the 14 age positively-correlated (schizophrenia – control increase with age) and 12 age negatively-correlated (schizophrenia – control decrease with age) genes of the significant GO term '*Regulation of kinase activity*' from the AT-schizophrenia/control NeuN+ cohort. (**F**) Example gene, WNK1 pairwise expression difference (schizophrenia – control) vs age (Pearson's *R*=0.73; p-value = 0.003). (**G**) Example gene, SFRP2 pairwise expression difference (schizophrenia – control) vs age (Pearson's *R*=–0.70; p-value = 0.005). (**H**) Example TF, EGR2 pairwise PPR difference (schizophrenia – control) vs age (Pearson's *R*=0.69; p-value = 0.0003) (**I**) Number of pairwise TF PPR and pairwise gene expression differences correlated with age in NeuN- nuclei from AF-schizophrenia/control pairs. (**J**) Number of pairwise TF PPR and pairwise gene expression differences correlated with age in NeuN- nuclei from AT-schizophrenia/control pairs. (**K**) Heatmap for the 5 age positively-correlated (schizophrenia – control increase with age) genes of the significant GO term '*Beta-catenin independent WNT signaling*' from the AT-schizophrenia/control NeuN- cohort.

marker of psychosis-like behavior (***González-Maeso et al., 2007***) – that we observed in older subjects (***Figure 7H***; ***Supplementary file 19***).

In NeuN- nuclei, 147 and 88 genes were identified with AF-schizophrenia/control and AT-schizophrenia/control expression difference vs age correlations of ≥0.60, respectively (***Figure 7I and J***; ***Supplementary file 19***). Enriched pathways in the AT-schizophrenia/control group included: '*Degradation of DVL*' (p-value $4.04\times10^{-5}$) and '*Beta-catenin independent WNT signaling*' (p-value $5.06\times10^{-4}$) (***Figure 7K***; ***Supplementary file 19***). Since dysfunctional WNT signaling is associated with several CNS disorders including Alzheimer's (***Wan et al., 2014***), together, these data also suggest that this positive correlation between NeuN- gene differences in AT-schizophrenia subjects/control pairs and age (***Figure 7K***; ***Supplementary file 19***) may be responsible for some of the negative effects of antipsychotic treatment on cognitive processes. We also identified 11 and 53 TFs correlated with age in the AF-schizophrenia/control and AT-schizophrenia/control cohorts, respectively (***Figure 7I and J***; ***Supplementary file 19***). However, as in NeuN+ nuclei, the effect of age became more evident in the AT-schizophrenia/control group with age-related adaptations in NeuN- TF-affected pathways that included '*Signaling by NOTCH*' (p-value $3.2\times10^{-4}$) (***Supplementary file 19***).

## Discussion

Understanding the molecular determinants involved in schizophrenia is critical for devising new treatment strategies and the discovery of the pathogenic mechanisms underlying this psychiatric condition. In this study, we combined low-input epigenomic and transcriptomic analysis to define how gene expression and TF regulation vary in schizophrenia subjects relative to controls and in response to antipsychotic treatment and aging. Our data provide evidence suggesting alterations in covalent histone modifications at different gene regions previously associated with schizophrenia risk, as well as additional genes involved in pathways related to immunological and neurodevelopmental processes. Whereas previous studies with postmortem human brain samples compared using indirect methods differences in chromatin accessibility between schizophrenia subjects and controls (***Bryois et al., 2018***), here we provide epigenomic signatures that distinguish between those observed in AF-schizophrenia subjects as well as alterations that denote previous treatment with antipsychotic medications.

We conducted a pairwise comparison between schizophrenia and matched controls, which is crucial to tease out treatment- or age-associated effects. A powerful feature of the *Taiji* framework is to allow the analysis of individual samples (***Zhang et al., 2019***). This integrative analysis of transcriptomic and epigenomic data at the systems level uncovered key regulators and important pathways based on their global importance in the genetic networks. We found that transcriptional mechanisms via novel pathways that had not been previously associated with schizophrenia show alterations in AF-schizophrenia subjects, and that these schizophrenia-linked pathways were statistically unaffected in the AT-schizophrenia group, consistent with a potential role of these epigenomic signatures in the clinical efficacy of antipsychotics. These changes appear to impact glutamatergic neurotransmission, IQGAP scaffold, autophagy, and ubiquitin B expression in neurons; and post NMDA receptor activation and long-term potentiation in glial cells. Additionally, our data highlight processes related to key pathways that may represent a consequence of antipsychotic medication, rather than a reversal of the molecular alterations observed in AF-schizophrenia subjects. These pathways suggest the existence of compensatory perturbations that emerge in response to repeated antipsychotic drug administration and

ultimately restrain their therapeutic effects (**Kurita et al., 2012**; **Ibi et al., 2017**). Among these, alterations in p53 activity were apparent as a consequence of antipsychotic treatment. Based on our ability to individually match schizophrenia and control pairs by age, we also revealed the intriguing observation that the effect of age on TF regulation of gene expression was significantly more pronounced in AT-schizophrenia subjects as compared to AF-schizophrenia subjects and controls.

Related to the effect of antipsychotic treatment, frontal cortex samples of schizophrenia subjects were divided into AF and AT based on postmortem toxicological analysis in both blood and when possible brain samples, which provides information about a longer retrospective drug-free period due to the high liposolubility of antipsychotic medications (**Voicu and Radulescu, 2009**). However, we cannot fully exclude the possibility of previous exposure to antipsychotic medications in the AF-schizophrenia group, and hence that the epigenetic alterations observed exclusively in the AF-schizophrenia group are a consequence of a potential period of decompensation, which typically occurs following voluntary treatment discontinuation (**Liu-Seifert et al., 2005**). It is also worth noting that our findings were established by examining the average characteristics of entire NeuN$^+$ and NeuN$^-$ fractions. Further studies of individual neuronal and glial cell subtypes may yield additional information on the role of cell-type-subpopulations (**Lau et al., 2020**; **Nagy et al., 2020**).

## Conclusion

Our ChIP-seq/RNA-seq study in postmortem brain samples from schizophrenia subjects and controls suggests cell-type specific epigenomic differences in individuals with schizophrenia, as well as cellular alterations in signaling pathways potentially involved in either the elimination of schizophrenia-related epigenomic alterations upon antipsychotic drug treatment or the antipsychotic-dependent modulation of alternative epigenetic pathways previously unaffected in the untreated schizophrenia cohort. Building on our data, future research could test the causal role of specific molecular pathways implicated in schizophrenia pathophysiology, as well as the therapeutic versus compensatory or negative side epigenomic outcomes induced by chronic treatment with antipsychotic medications.

# Materials and methods

## Post-mortem human brain tissue samples

Human brains were obtained during autopsies performed at the Basque Institute of Legal Medicine, Bilbao, Spain. The study was developed in compliance with policies of research and ethical review boards for post-mortem brain studies (**Arias-Diaz et al., 2013**). The Institutional Review Board (IRB) of the University of the Basque Country determined that approval was not required for this research in postmortem samples (M10/2018/283). According to legal requirements, samples were obtained by opting-out policy, absence of compensation for tissue donation and coded under reversible anonymization. Deaths were subjected to retrospective searching for previous medical diagnosis and treatment using examiner's information and records of hospitals and mental health centers. After searching for antemortem information was fulfilled, 29 subjects (Caucasian) who had met criteria of schizophrenia according to the Diagnostic and Statistical Manual of Mental Disorders (DSM-IV) (**American Psychiatric Association, 1994**) were selected. A toxicological screening for antipsychotics, other drugs, and ethanol was performed on blood samples collected at the time of death and, when possible, postmortem brain samples. The toxicological assays were performed at the National Institute of Toxicology, Madrid, Spain, using a variety of standard procedures including radioimmunoassay, enzymatic immunoassay, high-performance liquid chromatography, and gas chromatography-mass spectrometry. Controls (Caucasian) for the present study were chosen among the collected brains on the basis, whenever possible, of the following cumulative criteria: (i) negative medical information on the presence of neuropsychiatric disorders or drug abuse; (ii) appropriate sex, age, postmortem delay (time between death and autopsy), and freezing storage time to match each subject in the schizophrenia group; (iii) sudden and unexpected death (motor vehicle accidents); and (iv) toxicological screening for psychotropic drugs with negative results except for ethanol. Specimens of frontal cortex (Brodmann area 9) were dissected at autopsy (0.5–1 g tissue) on an ice-cooled surface and immediately stored at –80 °C until use. The schizophrenia subjects were divided into antipsychotic-free and antipsychotic-treated according to the presence or absence of antipsychotics in blood samples at the time of death. The definitive pairs of antipsychotic-free schizophrenics and respective matched

controls are shown in ***Supplementary file 1***, and the definitive pairs of atypical antipsychotic-treated schizophrenics and respective matched controls are shown in ***Supplementary file 2***. Presence or absence of antipsychotic medications was confirmed by toxicological analysis in postmortem brain samples of a selected group of schizophrenia subjects and controls (***Supplementary file 20***). Pairs of schizophrenia and matched controls were processed simultaneously and under the same experimental conditions. Tissue pH values were within a relatively narrow range (control subjects: 6.7±0.08; schizophrenic subjects: 6.6±0.06). Brain samples were also assayed for RIN (RNA integrity number) values using the Agilent 2100 Bioanalyzer (Applied Biosystems) – control subjects: 7.87±0.21; schizophrenic subjects: 7.61±0.32.

## Nuclei isolation and sorting via FACS

Nuclei isolation from frozen tissues (never fixed) of postmortem human brain samples was conducted using a published protocol (***Lake et al., 2016***) with some modifications. Frontal cortex samples from schizophrenic individuals and individually matched controls were always processed in the same batch. Briefly, all steps were conducted on ice, and all centrifugation was conducted at 4 °C. One piece of brain tissue (~300 mg) was placed in 3 ml of ice-cold nuclei extraction buffer (NEB) [0.32 M sucrose, 5 mM $CaCl_2$, 3 mM $Mg(Ac)_2$, 0.1 mM EDTA, 10 mM Tris-HCl, and 0.1%(v/v) Triton X-100] with freshly added 30 µl of protease inhibitor cocktail (PIC, Sigma-Aldrich), 3 µl of 100 mM phenylmethylsulfonyl fluoride (PMSF, Sigma-Aldrich) in isopropyl alcohol, 3 µl of 1 M dithiothreitol (DTT, Sigma-Aldrich), and 4.5 µl of recombinant ribonuclease (RNase) inhibitor (2313 A,Takara Bio). The tissue was homogenized in tissue grinder (D9063, Sigma-Aldrich). The homogenate was filtered with a 40 µm cell strainer (22-363-547, Thermo Fisher Scientific) and collected in a 15 ml centrifuge tube. The cell suspension was centrifuged at 1000 RCF at 4 °C for 10 min. The supernatant was discarded, and the pellet was resuspended in 0.5 ml of ice-cold NEB with freshly added 5 µl of PIC, 0.5 µl of PMSF, 0.5 µl of DTT, and 0.75 µl of RNase inhibitor. 500 µl of the sample was mixed with 750 µl of 50%(w/v) iodixanol (a mixture of 4 ml of OptiPrepTM gradient (Sigma-Aldrich) and 0.8 ml of diluent [150 mM KCl, 30 mM $MgCl_2$, and 120 mM Tris-HCl]). The mixture was centrifuged at 10,000 RCF at 4 °C for 20 min. Then, the supernatant was removed and 300 µl of 2%(w/v) normal goat serum (50062Z, Life technologies) in Dulbecco's PBS (DPBS, Life technologies) was added to resuspended the nuclei pellet. To label and separate $NeuN^+$ and $NeuN^-$ fractions, 6 µl of 2 ng/ml anti-NeuN antibody conjugated with Alexa 488 (MAB377X, Millipore) in DPBS was added into the nuclei suspension. The suspension was mixed well and incubated at 4 °C for 1 hr on a rotator mixer (Labnet). After incubation, the sample was sorted into $NeuN^+$ and $NeuN^-$ populations using a BD FACSARIATM Flow Cytometer (BD Biosciences). 400 µl of sorted nuclei suspension ($NeuN^+$ or $NeuN^-$), containing ~50,000 nuclei (for conducting ChIP-seq and input libraries), was added into 600 µl of ice-cold PBS. 200 µl of 1.8 M sucrose solution, 5 µl of 1 M $CaCl_2$, and 3 µl of 1 M $Mg(Ac)_2$ were added into the mixture. The solution was mixed well and incubated on ice for 15 min. Then, the sample was centrifuged at 1800 RCF at 4 °C for 15 min. The supernatant was discarded and the pellet was resuspended in 110 µl of PBS with freshly added 1.1 µl of PIC and 1.1 µl of PMSF and stored on ice until use.

## Construction of ChIP-seq libraries

Chromatin fragments were prepared by using micrococcal nuclease (MNase) to digest sorted and concentrated nuclei ($NeuN^+$/$NeuN^-$) following a published protocol (***Zhu et al., 2019***). 54 µl of chromatin fragments (from 10,000 nuclei) was used in each ChIP assay for producing two replicate libraries. Chromatin immunoprecipitation was carried out using multiplexed MOWChIP assay (***Zhu et al., 2019***) with anti-H3K4me3 (39159, Active Motif) and anti-H3K27ac (39135, Active Motif) antibody. ChIP-seq libraries were prepared using Accel-NGS 2 S Plus DNA Library kit (Swift Biosciences) from the purified immunoprecipitated DNA. The library preparation was conducted without knowledge of the brain sample or the type of histone mark. Minor modification was made to the manufacturer's procedures as detailed below. In the amplification step, instead of adding 10 µl of low EDTA TE buffer into each reactoin, we added the mixture of 7.5 µl of low EDTA TE buffer and 2.5 µl of 20 X Evagreen dye to monitor and quantify PCR amplification. The reaction was stopped when the sample's fluorescence intensity increased by 3000 relative fluorescence units (RFU). Then, 50 µl of the mixture after PCR amplification was transferred into an Eppendorf tube and mixed with 37.5 µl of SPRI select beads.

After 5 min incubation at room temperature, the beads went through a cleanup procedure with 80% ethanol. In the end, the DNA library was eluted from the beads into 7 µl of low EDTA TE buffer.

## Construction of RNA-seq libraries

100 µl of sorted nuclei suspension ($NeuN^+$ or $NeuN^-$) from brain tissue, containing ~12,000 nuclei for producing two replicate libraries, was used for RNA extraction by using the RNeasy Mini Kit (74104, Qiagen) and RNase-Free DNase Set (79254, Qiagen), following the manufacturer's instruction. Half of the extracted mRNA (from 6000 nuclei) in 30 µl volume was concentrated by ethanol precipitation and resuspended in 4.6 µl of RNase-free water. mRNA-seq libraries were prepared using Smart-seq2 (***Picelli et al., 2013***) and a Nextera XT DNA Library Preparation kit (FC-131–1024, Illumina) following the protocol and the manufacturer's instructions with minor modification. ~2 ng of mRNA (in 4.6 µl of water) was mixed with 2 µl of 100 mM oligo-dT primer and 2 µl of 10 mM dNTP mix. After being denatured at 72 °C for 3 min, the mRNA solution was immediately placed on ice. Then, 11.4 µl of reverse transcript mix [1 µl of SuperScript II reverse transcriptase (200 U/ml), 0.5 µl of RNAse inhibitor (40 U/ml), 4 µl of Superscript II first-strand buffer, 1 µl of DTT (100 mM), 4 µl of 5 M Betaine, 0.12 µl of 1 M $MgCl_2$, 0.2 µl of TSO (100 mM), 0.58 µl of nuclease-free water] was mixed with the mRNA solution and the mixture was incubated at 42 °C for 90 min, followed by 10 cycles of (50 °C for 2 min, 42 °C for 2 min). The reaction was finally inactivated at 70 °C for 15 min. 20 µl of first-strand mixture was then mixed with 25 µl of KAPA HiFi HotStart ReadyMix, 0.5 µl of (100 mM) IS PCR primers, 0.5 µl of Evagreen dye, and 4 µl of nuclease-free water. Generated complementary DNA (cDNA) was amplified by incubated at 98 °C for 1 min, followed by 9–11 cycles of (98 °C 15 s, 67 °C 30 s, 72 °C 6 min). After PCR amplification, 50 µl of PCR mix was purified by using 50 µl of SPRIselect beads. ~600 pg of purified cDNA was used for Nextera XT library preparation. ChIP-seq and RNA-seq library fragment size were measured by using high sensitivity DNA analysis kit (5067–4626, Agilent) on a TapeStation system (2200, Agilent). After this, 18–22 ChIP-seq and RNA-seq libraries were randomly pooled together. Around 15 and 11 million reads were allocated to each ChIP-seq and RNA-seq library, respectively. The concentration of each library was examined by a KAPA library quantification kit (KK4809, Kapa Biosystems), and then the quantified libraries were pooled at 10 nM. The libraries were sequenced by Illumina HiSeq 4000 with single-end 50-nt read.

## ChIP-seq data processing

Raw ChIP-seq reads and input data, were mapped to human genome (GRCh38) using Bowtie2 (2.2.5). Peaks were called using MACS2 (2.2.7.1) using a q-value cutoff of 0.05 for the narrow marks (H3K4me3 and H3K27ac).

## RNA-seq data processing

The human genome (GRCh38) and comprehensive gene annotation were obtained from GENCODE (v29). Quality control of RNA-seq reads including per sequence GC and adapter content was assessed with FastQC. Reads were mapped with STAR (2.7.0 f) with soft-clipping (average of 73.8% (+/-0.08%) reads uniquely mapped for neurons and 69.0% (+/-0.99%) reads for glia) and quantified with feature-Counts (v2.0.1) using the default parameters.

## Differential analysis for ChIP-seq data

The peaks were called using MACS2 (***Zhang et al., 2008***). The peaks with q-value <0.05 were taken as input for diffBind R package. We first created cell-type-specific consensus peak sets using Diffbind for neurons and glia separately. Using the function of dba.peakset in diffbind (hg38_blacklist_remove, consensus = DBA_REPLICATE, minOverlap = 2), we detected the 'high-confidence' peaks if they were identified in both of the technical replicates of the sample (n=58) in either schizophrenia or control groups and then the 'high-confidence' peak sets from each sample of the two groups were combined into a master set of consensus peaks for analysis. The raw read counts were extracted using the function of dba.count (hg38_blacklist_remove, summits = FALSE, peaks = consensus_peaks,filter = 1, bScaleControl = TRUE, minCount = 1, score = DBA_SCORE_TMM_MINUS_FULL) in diffBind, and the peaks with less than 20 reads in over 50% of the samples were removed before differential analysis. DESeq2 R package was used to perform the differential peaks analysis based on the TMM normalized reads to identify differential peaks between schizophrenia and control cohort (adjusted p-value

<0.05). The p-values were adjusted by performing a standard Bonferroni correction. The following covariates were regressed out: demographic covariates (age at death, sex, PMD, and diagnosis) and technical covariates (align rate, unique rate, FRiP, NSC, RSC, the number of identified peaks, and PBC) by correlating the top 6 principal components with these covariates. We annotated enhancers (defined as identified H3K27ac peaks that have no overlap regions with promoters) to genes using published Hi-C data on neurons and glia (***Hu et al., 2021***) when possible and the rest of the enhancers were associated with their nearest genes. We annotated H3K4me3 peaks to genes when they overlapped with the promoter regions.

## ChIP-seq annotation and functional enrichment

GREAT analysis (http://great.standford.edu) was performed on differential peaks using the whole genome as background and default basal extension from 5 kb upstream to 1 kb downstream of the TSS. Significantly enriched Gene Ontology biological processes were identified using the Panther Classification tools using a hypergeometric test.

## Differential analysis for RNA-seq data

We analyzed the bulk RNA-seq data of 29 schizophrenia subjects and 29 controls. The initial step involved filtering out genes with low read counts (less than 20 reads in over 50% of samples). The analysis then employed a two-step method to estimate the technical and biological noise. The first step was identifying the top 10 principal components (PCs) of the dataset. Subsequently, the correlation between each PC and various experimental (alignment rate, unique rate, exon percentage, number of unique mapped reads) and demographic (sex, age at death, PMD, antemortem diagnosis) factors was calculated. Covariates with high correlation to the PCs were included in the analysis to minimize their impact. The analysis was conducted using the 'DESeq2' software package, and genes with a false discovery rate (FDR) below 0.05 were identified as differentially expressed.

## ChIP-qPCR assays

After nuclei extraction, MNase digestion and MOWChIP assays (see above), ChIP DNA was eluted to 10 µl of low EDTA TE buffer. 1 µl of ChIP DNA solution was used for qPCR assays with each primer set. The following qPCR primer pairs were used:

***Figure 2—figure supplement 1A***: AGG GAC CTG GAA CAT CTT TG (F); CAT CAT CCT CAG AAG GAG TCT G (R)
***Figure 2—figure supplement 1B***: TGG AGA TAG GTG GAT GTT AAG C (F); CCA TAT TGA CCC TGG GCT ATT (R)
***Figure 2—figure supplement 1C***: ATG CCA ATT AGG CTA TAG ATG CT (F); CTT AAC AGG GCA CTC TCA GTA AT (R)
***Figure 2—figure supplement 1D***: AAA GAG CAA GCA GGG ACT T (F); GAT GTA ATA ACG TGG GAG AGA GG (R)
***Figure 2—figure supplement 1E*** AGG AGT GGA TAC AGG GAG ATT AG (F); TGT GTA TTC TGT GTC TGG CTT T (R)
***Figure 2—figure supplement 1F***: ACC AAC GAA TAC CCT GCT TT (F); AAG GCC TGG CAA CCT TAA T (R)

The following common negative primer set was used in all samples, against which the enrichment of each positive set was calculated:

GCA GAA CCT AGT TCC TCC TTC AAC (F); AGT CAT CCC TTC CTA CAG ACT GAG A (R)

qPCR primer sets were ordered from IDT, made to lab-ready formulation (100 µM in low EDTA TE buffer). Ready to use stocks of primer sets were made by combining 10 µl each of both forward and reverse primers of the same set with 80 µl of low EDTA TE buffer. 10 µl of iQ SYBR Green Supermix, 1.6 µl of primer stock, 1 µl of ChIP DNA, and 7.4 µl of ultrapure water were added to each qPCR well. Reaction was conducted on a CFX96 real-time PCR machine (Bio-Rad) with C1000 thermal cycler base. All PCR assays were performed using the following thermal cycling profile: 95 °C for 10 min followed by 40 cycles of (95 °C for 15 s, 58 °C for 40 s, 72 °C for 30 s). Relative fold enrichment of each positive

primer (P) against the common negative primer (N) was calculated using the following equation: Enrichment = $2^{Cq(N) - Cq(P)}$.

## Taiji pipeline

Active regulatory elements were first identified via the overlap of high confidence peaks from H3K27ac with known gene promoter regions (4kbp upstream and 1kbp downstream of the transcription start sites). The distal H3K27ac peaks were assigned to active promoters using the unsupervised learning method EpiTensor, and assigned as an enhancer-promoter interaction if one locus overlapped with the distal peak and the other locus in the pair overlapped with a known promoter. Putative TF binding motifs were curated from the CIS-BP database (**Weirauch et al., 2014**). Using FIMO's algorithm (**Grant et al., 2011**). TFs were identified as having binding sites within 150 bp regions centered around H3K27ac peak summits. 58 unique $NeuN^+$ (29 schizophrenia and 29 control) and 58 unique $NeuN^-$ (29 schizophrenia and 29 control) network topologies were thus constructed by forming directed edges between TF and their regulatees, if the TF had a predicted binding site in the gene's promoter or linked enhancer.

## Personalized PageRank (PPR)

The PPR algorithm was run to measure the global influence of each node. To initialize the networks, node weights were initialized separately in each cell-type *i*, where a gene's relative expression level is a z-score transformation of its absolute expression, $z_i$ and the node weight for this gene in cell type *i* is then given by $e^{z_i}$. Edge weights were determined according to the expression level of the parent node TF and the pooled H3K27ac ChIP-seq peak intensity (strength of the TF-gene association) as previously reported (**Zhang et al., 2019**). The directionality of the topological edges was reversed and the normalized node weights were then used as the seed vector for the PPR calculation. Post convergence, edge directionality was re-reversed.

## EpiSig analysis

To integrate H3K27ac, H3K4me3, and RNA-seq data from two cell types across the postmortem frontal cortex samples from schizophrenia subjects and controls, EpiSig was employed (**Ai et al., 2018**). This algorithm detects the significant signals from sequencing data in 5 kb bins across the whole genome, and then clusters the regions based on the similar epigenomic profiles across all samples.

## EpiSig differential enrichment analysis

A hypergeometric test was applied to all *EpiSig* clusters to assess the enrichment of differential H3K27ac and H3K4me3 peaks and differentially expressed genes. Clusters with FDR <0.05 were selected and then ranked according to the number of overlapping peaks for each mark. Peaks were then mapped to genes using GREAT with default settings.

## Age correlation analysis

Raw expression, pairwise expression and pairwise TF PPR age correlations were calculated using the Pearson R correlation. Significance was assessed by calculating p-values for the Pearson R correlations using the t-distribution with n-2 degrees of freedom for the respective cohort.


# Acknowledgements

CL dedicates this paper to his elder brother Lv Wei who passed away in 2023 after a 30 year battle with schizophrenia. The authors thank the staff members of the Basque Institute of Legal Medicine for their cooperation in the study. National Institutes of Health R01MH084894 (JG-M), R01MH111940 (JG-M), R01GM143940 (CL), R01GM141096 (CL), R01HG009626 (WW), R01AI50282 (WW), Eusko Jaurlaritza IT1211-19 (JJM) and IT-1512/22 (LFC), and VCU Presidential Request Fund (JG-M).


# Additional information

### Competing interests

J Javier Meana: received unrestricted funds from Janssen. Javier González-Maeso: has sponsored research contracts with Terran Biosciences and Gonogo Solutions. The other authors declare that no competing interests exist.


## Funding

| Funder | Grant reference number | Author |
|---|---|---|
| National Institutes of Health | R01MH084894 | Javier González-Maeso |
| National Institutes of Health | R01MH111940 | Javier González-Maeso |
| National Institutes of Health | R01GM143940 | Chang Lu |
| National Institutes of Health | R01HG009626 | Wei Wang |
| National Institutes of Health | R01AI50282 | Wei Wang |
| Virginia Commonwealth University | VCU Presidential Request Fund | Javier González-Maeso |
| National Institutes of Health | R01GM141096 | Chang Lu |
| Eusko Jaurlaritza | IT1211-19 | J Javier Meana |
| Eusko Jaurlaritza | IT-1512/22 | Luis F Callado |

The funders had no role in study design, data collection and interpretation, or the decision to submit the work for publication.


## Author contributions

Bohan Zhu, Richard I Ainsworth, Data curation, Formal analysis, Investigation, Methodology, Writing – original draft; Zengmiao Wang, Zhengzhi Liu, Formal analysis, Investigation; Salvador Sierra, Investigation, Methodology; Chengyu Deng, Investigation; Luis F Callado, J Javier Meana, Resources, Investigation; Wei Wang, Chang Lu, Conceptualization, Resources, Data curation, Software, Formal analysis, Supervision, Funding acquisition, Methodology, Writing – review and editing; Javier González-Maeso, Conceptualization, Resources, Data curation, Formal analysis, Supervision, Funding acquisition, Methodology, Writing – review and editing


## Author ORCIDs

Bohan Zhu http://orcid.org/0009-0003-9823-8630
Richard I Ainsworth http://orcid.org/0000-0002-3350-5692
Luis F Callado https://orcid.org/0000-0001-9941-012X
J Javier Meana https://orcid.org/0000-0002-7913-6714
Wei Wang http://orcid.org/0000-0003-4377-5060
Chang Lu http://orcid.org/0000-0003-0181-5888
Javier González-Maeso https://orcid.org/0000-0003-3105-3204


## Ethics

Human brains were obtained during autopsies performed at the Basque Institute of Legal Medicine, Bilbao, Spain. The study was developed in compliance with policies of research and ethical review boards for post-mortem brain studies (Arias-Diaz et al 2013). The Institutional Review Board (IRB) of the University of the Basque Country determined that approval was not required for this research in postmortem samples (M10/2018/283). According to legal requirements, samples were obtained by opting-out policy, absence of compensation for tissue donation and coded under reversible anonymization.



Reviewer #1 (Public Review): https://doi.org/10.7554/eLife.92393.3.sa1
Author response https://doi.org/10.7554/eLife.92393.3.sa2

## Additional files

### Supplementary files

- Supplementary file 1. Demographic information of antipsychotic-free (AF)-schizophrenia subjects and controls.
- Supplementary file 2. Demographic information of antipsychotic-treated (AT)-schizophrenia subjects and controls.
- Supplementary file 3. Quality control measurements on ChIP-seq datasets.
- Supplementary file 4. Quality control measurements on RNA-seq datasets.
- Supplementary file 5. Transcription frequency in transcripts per kilobase million (TPM) on key marker genes for neuronal and glial nuclei samples from control and schizophrenia subjects.
- Supplementary file 6. Differential histone modification peaks and differentially expressed genes (DEGs) obtained by comparing schizophrenia and controls.
- Supplementary file 7. Enrichment of various genome-wide association studies (GWAS) traits in differential enhancers and promoters.
- Supplementary file 8. Overlap of differentially expressed genes (DEGs) identified in our study and a previous snRNA-seq study.
- Supplementary file 9. Genes identified via differential analyses of enhancers, promoters, and RNA by comparing schizophrenia and controls.
- Supplementary file 10. *EpiSig* analysis in NeuN-positive ($NeuN^+$) nuclei from the frontal cortex of schizophrenia subjects and controls.
- Supplementary file 11. *EpiSig* analysis in NeuN-negative ($NeuN^-$) nuclei from the frontal cortex of schizophrenia subjects and controls.
- Supplementary file 12. *Taiji* analysis in NeuN-positive ($NeuN^+$) and NeuN-negative ($NeuN^-$) nuclei from the frontal cortex of schizophrenia subjects and controls.
- Supplementary file 13. *Taiji* analysis in NeuN-positive ($NeuN^+$)and NeuN-negative ($NeuN^-$) nuclei from the frontal cortex of antipsychotic-free (AF)-schizophrenia subjects, antipsychotic-treated (AT)-schizophrenia subjects, and controls.
- Supplementary file 14. *EpiSig* analysis in NeuN-positive ($NeuN^+$) nuclei from the frontal cortex of antipsychotic-free (AF)-schizophrenia subjects and controls.
- Supplementary file 15. Differential H3K27ac/H3K4me3 peaks and DEGs obtained by comparing antipsychotic-free (AF)-schizophrenia and antipsychotic-treated (AT)-schizophrenia with their respective matched controls.
- Supplementary file 16. Genes altered in antipsychotic-free (AF-) but not antipsychotic-treated (AT)-schizophrenics, and those altered in AT- but not AF-schizophrenics. The lists of genes were obtained by differential analyses of enhancers, promoters, and RNA in AF-schizophrenia, AT-schizophrenia, and their respective controls.
- Supplementary file 17. *EpiSig* analysis in NeuN-positive ($NeuN^+$) nuclei from the frontal cortex of antipsychotic-treated (AT)-schizophrenia subjects and controls.
- Supplementary file 18. *EpiSig* analysis in NeuN-negative ($NeuN^-$) nuclei from the frontal cortex of antipsychotic-treated (AT)-schizophrenia subjects and controls.
- Supplementary file 19. Effect of age on chromatin organization in the frontal cortex of AF-schizophrenia subjects, AT-schizophrenia subjects, and controls.
- Supplementary file 20. Toxicological analysis in postmortem human brain samples.
- MDAR checklist

### Data availability

The raw ChIP-seq and RNA-seq data were deposited in dbGaP under accession number phs002487.v1.p1. The processed data can be accessed via Gene Expression Omnibus (GEO) under accession number GSE174407. The code used for the analysis shown in this manuscript was deposited on GitHub (copy archived at ***Chang Lu lab, 2023***).

The following datasets were generated:

| Author(s) | Year | Dataset title | Dataset URL | Database and Identifier |
|---|---|---|---|---|
| Zhu B, Deng C, Lu C | 2021 | Antipsychotic-induced epigenomic reorganization in frontal cortex of individuals with schizophrenia | http://www.ncbi.nlm.nih.gov/geo/query/acc.cgi?acc=GSE174407 | NCBI Gene Expression Omnibus, GSE174407 |
| Zhu B, Ainsworth RI | 2023 | Epigenomes and Transcriptomes of Brain Samples of Schizophrenics and Controls | https://www.ncbi.nlm.nih.gov/projects/gap/cgi-bin/study.cgi?study_id=phs002487.v1.p1 | dbGaP, phs002487.v1.p1 |

The following previously published dataset was used:

| Author(s) | Year | Dataset title | Dataset URL | Database and Identifier |
|---|---|---|---|---|
| Girdhar et al | 2018 | PsychENCODE Knowledge Portal | https://doi.org/10.7303/syn4566010 | Synapse, 10.7303/syn4566010 |

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
