## [Editor Report · eLife assessment]

The study by Zhu et al. provides **important** insights into cell-specific genome-wide histone modifications in the frontal cortex of individuals with schizophrenia, as well as shedding light on the role of age and antipsychotic treatment in these associations. The evidence supporting the conclusions is **solid**.

---

## [Referee Report · Reviewer #1 (Public Review)]

Zhu, et al present a genome-wide histone modification analysis comparing patients with schizophrenia (on or off antipsychotics) to non-psychiatric controls. The authors performed analyses across the dorsolateral prefrontal cortex and tested for enrichment of nearby genes and pathways. The authors performed analysis measuring the effect of age on the epigenomic landscape as well. This paper provides a unique resource around SCZ and its epigenetic correlates, and some potentially intriguing findings in the antipsychotic response dataset.

Comments on revised version:

The authors have adequately responded to my review comments.

---

## [Author Response]

The following is the authors’ response to the original reviews.

**Public Reviews:**

**Reviewer #1 (Public Review):**
Zhu, et al present a genome-wide histone modification analysis comparing patients with schizophrenia (on or off antipsychotics) to non-psychiatric controls. The authors performed analyses across the dorsolateral prefrontal cortex and tested for enrichment of nearby genes and pathways. The authors performed an analysis measuring the effect of age on the epigenomic landscape as well. While this paper provides a unique resource around SCZ and its epigenetic correlates, and some potentially intriguing findings in the antipsychotic response dataset there were some potential missed opportunities - related to the integration of outside datasets and genotypes that could have strengthened the results and novelty of the paper.Major Comments(1) Is there genotype data available for this cohort of donors or can it be generated? This would open several novel avenues of investigation for the authors. First the authors can test for enrichment of heritability for SCZ or even highly comorbid disorders such as bipolar. Second, it would allow the authors to directly measure the genetic regulation of histone markers by calculating QTLs (in this case histone hQTLs). The authors assert that although interesting, ATACseq approach does not provide the same chromatin state information as histone mods mapped by ChiP. Why do the authors not test this? There are several ATACseq datasets available for SCZ [https://pubmed.ncbi.nlm.nih.gov/30087329/]and an additional genomic overlap could help tease apart genetic regulation of the changes observed.

As detailed in our Methods section, brain samples have previous medical diagnosis, treatment record, and toxicological screening. Unfortunately, there was no genotype information on our brain sample collection. However, we examined overlap of differential enhancer and promoter peaks with genetic variants using linkage disequilibrium score regression (Fig. S10). Additionally, to assess agreement with the literature, we compared DEGs identified in our study with a previous snRNA-seq study in postmortem prefrontal cortex of schizophrenics and controls (Table S7).

Repressive histone marks tend to provide different information than ATAC-seq data.However, we examined only activating marks in this study. Thus, the sentence in the Introduction mentioning that “ATAC-seq approach does not provide the same chromatin state information as histone modifications mapped by chromatin immunoprecipitation sequencing (ChIP-seq) assays do” has been removed.

(2) Can the authors theorize why their analysis found significant effects for H3K27Ac for antipsychotic use when a recent epigenomic study of SCZ using a larger cohort of samples and including the same histone modifications did not [https://pubmed.ncbi.nlm.nih.gov/30038276/]? Given the lower n and lower number of cells in this group, it would be helpful if the authors could speculate on why they see this. Do the authors know if there is any overlap with the Girdhar study donors or if there are other phenotypic differences that could account for this?

As mentioned in the Methods sections, three strengths of this brain bank include (i) inclusion of samples of schizophrenia subjects with antemortem diagnosis (i.e., based on clinical histories) and not with postmortem diagnosis (i.e., based on interviews with relatives and friends – a diagnostic approach used by many brain banks worldwide but with important limitations, see here: PMID: 15607306), (ii) inclusion of control subjects individually matched by sex, age and PMD, and (iii) our possibility to test the presence or absence of antipsychotic medications in blood samples as an independent experimental variable. This allowed us to obtained novel and statistically valid conclusions related to cell-type epigenetic alterations in the frontal cortex of schizophrenia subjects, and the impact of age and antipsychotic treatment on chromatin organization.

There is no overlap with Girdhar study donors.

(3) The reviewer is concerned about the low concordance between bulk nucleiRNA-seq and single-cell RNA-seq for SCZ (236 of 802 DEGs in NeuN+ and 63 of 1043 NEuN-). While it is not surprising for different cohorts to have different sets of DEGs these seem to be vastly different. Was there a particular cell type(s) that enriched for the authors' DEGs in the single-cell dataset? Do the authors know if any donors overlapped between these cohorts?

This overlap is acceptable considering that these are datasets originated from an entirely distinct cohort of postmortem human brain samples.

(4) Functional enrichment analyses: details are not provided by the authors and should be added. The authors need to consider (a) providing a gene universe, ie only considering the sets of genes with nearby H3K4me3/ H3K27ac levels, to such pathway tools, and (b) should take into account the fact that some genes have many more peaks with data. There are known biases in seemingly just using the best p-value per gene in other epigenetic analysis (ie. DNA methylation data) and software is available to run correct analyses: https://pubmed.ncbi.nlm.nih.gov/23732277.

GREAT was used to map differential peak loci to target genes using the whole genome as the background set and default basal extension as per Nord et al.http://dx.doi.org/10.1016/j.cell.2013.11.033. We argue that it is more biologically relevant than comparing against an artificially selected background. These gene sets were then passed to Panther for Gene Ontology enrichment analysis as per Liu et al.10.1186/s12940-015-0052-5.

Additional details are provided in Materials and Methods section:

ChIP-seq annotation and functional enrichment

GREAT analysis (http://great.standford.edu) was performed on differential peaks using the whole genome as background and default basal extension from 5kb upstream to 1kb downstream of the TSS.

Significantly enriched Gene Ontology biological processes were identified using the Panther Classification tools using a hypergeometric test.

**Reviewer #2 (Public Review):**
The manuscript by Zhu has generated ChIP-seq and RNA-seq data from sizeable cohorts of SCZ patient samples and controls. The samples include 15 AF-SCZ samples and 15 controls, as well as 14 AT-SCZ samples and 14 controls. The genomics data was generated using techniques optimized for low-input samples: MOWChIP-seq and SMART-seq2 for histone profiles and transcriptome, respectively. The study has generated a significant data resource for the investigation of epigenomic alterations in SCZ. I am not convinced that the hierarchical pairwise design - first comparing AF-SCZ and AT-SCZ with their corresponding controls and secondarily contrasting the two comparisons is fully justified. The authors should repeat the statistical analysis by modeling all three groups simultaneously with an interaction effect for treatment or directly compare AF-SCZ to AT-SCZ groups and evaluate if the main conclusions remain supported.Major comments(1) The manuscript did not discuss (mention) the quality control of RNA-seq data shown in Fig. 1B. The color scheme choice for the heatmap visualization did not provide a quantitative presentation of the specificity of the RNA-seq data. I would recommend using bar plots to present the results more quantitatively.

QC of raw RNA-seq data including per sequence GC and adapter content was assessed with FastQC. Reads underwent soft-clipping during STAR alignment with on average 73.8% (+/- 0.08%) reads for neurons and 69.0% (+/- 0.99%) reads for glia being uniquely mapped. A new supplementary figure (Figure S5) has been included to show four bar plots representing the expression values more quantitatively.

These details are now provided in the RNA-seq data processing part of the Materials and Methods section:

RNA-seq data processing

The human genome (GRCh38) and comprehensive gene annotation were obtained from GENCODE (v29). Quality control of RNA-seq reads including per sequence GC and adapter content was assessed with FastQC. Reads were mapped with STAR (2.7.0f) with soft-clipping (average of 73.8% (+/- 0.08%) reads uniquely mapped for neurons and 69.0% (+/- 0.99%) reads for glia) and quantified with featureCounts (v2.0.1) using the default parameters.

(2) How does the specificity of this RNA-seq dataset compare to previous studies using a similar NeuN sorting strategy?As mentioned in the Results section, highly significant (median p-value = 6 ´ 10-7) pairwise differences in molecular marker expression were observed for all markers ranging from mature, functional and synaptic neuron markers to astrocyte, oligodendrocyte and microglial markers (Figure 1B; Figures S4 and S5; Table S5). This confirms neuronal and non-neuronal cell-type identities in the NeuN+ and NeuN- nuclei samples, respectively.(3) I appreciate the effort to assess the ChIP-seq data quality using phantompeakqualtools. However, prior knowledge/experience with this tool is required to fully understand the QC results. The authors should additionally provide browser shots at different scales for key neuronal/glial genes, so readers can have a more direct assessment of data quality, such as the enrichment of H3K4me3 at promoters (but not elsewhere), and H3K27ac at promoters and enhancers. Existing browser views, such as Fig. 2B are too zoomed out for assessing the data quality.

A new Fig 2B has been generated with a magnified view for clearer examination.

(4) The pairwise regression model should be explicitly reported in methods.

Additional details are included in the Methods section:

Differential analysis for RNA-seq data

We analyzed the bulk RNA-seq data of 29 schizophrenia subjects and 29 controls. The initial step involved filtering out genes with low read counts (less than 20 reads in over 50% of samples). The analysis then employed a two-step method to estimate the technical and biological noise. The first step was identifying the top 10 principal components (PCs) of the dataset. Subsequently, the correlation between each PC and various experimental (alignment rate, unique rate, exon percentage, number of unique mapped reads) and demographic (sex, age at death, PMD, antemortem diagnosis) factors was calculated. Covariates with high correlation to the PCs were included in the analysis to minimize their impact. The analysis was conducted using the 'DESeq2' software package, and genes with a false discovery rate (FDR) below 0.05 were identified as differentially expressed.

(5) The statistical strategy to compare AF-SCZ and AT-SCZ to their corresponding control groups was unjustified. Why not model all three groups simultaneously with an interaction effect for treatment or directly compare AF-SCZ to AT-SCZ groups? If the manuscript argues that the antipsychotic effect is the main novelty, why not directly compare AF-SCZ and AT-SCZ?

This is an important point. As mentioned above, one of the main strengths of our experimental design is that schizophrenia subjects and controls were individually matched by sex and age and (if possible) postmortem delay and freezing storage time. Our study is also among the first to report the potential impact of antipsychotic treatment on chromatin organization using postmortem human brain samples. Because of this individual matching method, we only compared schizophrenia subjects (either antipsychotic-free or antipsychotic-treated) with their respective individually matched controls. This experimental design is supported by our previous publications with postmortem human brain samples (PMID: 36100039; PMID: 28783139; PMID: 26758213; PMID: 23129762; PMID: 22864611; PMID: 18297054). The rationale behind this experimental design – as well as potential limitations particularly related to the division of the schizophrenia group in antipsychotic-free and antipsychotic-treated – is mentioned in the Discussion:

Related to the effect of antipsychotic treatment, frontal cortex samples of schizophrenia subjects were divided into AF and AT based on postmortem toxicological analysis in both blood and when possible brain samples, which provides information about a longer retrospective drug-free period due to the high liposolubility of antipsychotic medications (Voicu and Radulescu, 2009). However, we cannot fully exclude the possibility of previous exposure to antipsychotic medications in the AF-schizophrenia group, and hence that the epigenetic alterations observed exclusively in the AF-schizophrenia group are a consequence of a potential period of decompensation, which typically occurs following voluntary treatment discontinuation (Liu-Seifert et al., 2005).

It is also worth mentioning here that data were analyzed both at the cohort level, as well as at an individual level (schizophrenia/cohort pairs). This is mentioned in the manuscript:

It should be noted that in the differential analyses here, the schizophrenia subjects (whether AF or AT) and their controls were compared at the cohort level, while matched schizophrenia/control pairs were examined individually in the TF-based analyses.

(6) The method of pairwise comparison to corresponding control groups, then further comparing the pairwise results opens the study to a number of statistical vulnerabilities. For example, on page 12, the studies identified 166 DEGs between AF and control, and 1273 DEGs between AT and control. Instead of implicating a greater amount of difference between AT and control, such a result can often be driven by differences in between-group variance, rather than between-group means, that is, are the SCZ-AF and SCZ-treated effect size magnitudes and directionalities similar (but the treated group has lower variance) or are the two groups truly different in terms of means? The result in Fig. 5A suggests effect sizes for the two comparisons (AF-Ctrl and AT-Ctrl) are similar but have lower variability in the treated group.

For a discussion regarding our approach, which involves a pairwise comparison, see above.

(7) The pairwise comparison further raised the possibility the results were driven by the difference in the two control cohorts rather than the two SCZ cohorts.

We clearly show that age is an important independent factor (Fig 7). Since controls are individually matched by sex and age, this limits the validity of the comparison among the two cohort groups including subjects of different age (see Tables S1 and S2).

**Recommendations for the authors:**

**Reviewer #1 (Recommendations For The Authors):**
Minor Comments(1) Why not mention what histone modifications you measured by Chip-seq in the abstract? A certainly minor point but I felt I read for quite a while before I got to that point in the intro.

The two histone marks are now mentioned in the abstract.

(2) There are several places in the introduction where improper grammar is utilized and this should be edited.

Introduction has been edited.

(3) Related to major comments, how many donors overlapped with the PsychENCODE, CommonMind papers?

Our datasets were generated from an entirely distinct cohort of postmortem human brain samples. Our postmortem sample collection does not overlap with postmortem samples included in PsychENCODE and/or CommonMind publications.

(4) Since studies have already measured H3K4me3 and H3K27ac in the SCZ prefrontal cortex, why didn't the authors consider measuring changes in a related repressive marker? This is not to suggest the authors should do that now, but additional comments about other markers would help provide context for this analysis and point toward potential future studies.

This is an interesting question and will be the goal of our future investigation.